# A signal capture and proofreading mechanism for the KDEL-receptor explains selectivity and dynamic range in ER retrieval

**Andreas Gerondopoulos[†], Philipp Bräuer[†], Tomoaki Sobajima, Zhiyi Wu, Joanne L Parker, Philip C Biggin, Francis A Barr\*, Simon Newstead\***

Department of Biochemistry, University of Oxford, Oxford, United Kingdom

**Abstract** ER proteins of widely differing abundance are retrieved from the Golgi by the KDEL-receptor. Abundant ER proteins tend to have KDEL rather than HDEL signals, whereas ADEL and DDEL are not used in most organisms. Here, we explore the mechanism of selective retrieval signal capture by the KDEL-receptor and how HDEL binds with 10-fold higher affinity than KDEL. Our results show the carboxyl-terminus of the retrieval signal moves along a ladder of arginine residues as it enters the binding pocket of the receptor. Gatekeeper residues D50 and E117 at the entrance of this pocket exclude ADEL and DDEL sequences. D50N/E117Q mutation of human KDEL-receptors changes the selectivity to ADEL and DDEL. However, further analysis of HDEL, KDEL, and RDEL-bound receptor structures shows that affinity differences are explained by interactions between the variable −4 H/K/R position of the signal and W120, rather than D50 or E117. Together, these findings explain KDEL-receptor selectivity, and how signal variants increase dynamic range to support efficient ER retrieval of low and high abundance proteins.

**\*For correspondence:**
francis.barr@bioch.ox.ac.uk (FAB);
simon.newstead@bioch.ox.ac.uk (SN)

[†]These authors contributed equally to this work

**Competing interests:** The authors declare that no competing interests exist.

## Introduction

Stable maintenance of the luminal composition of the endoplasmic reticulum (ER) is necessary for the function of the secretory pathway (*Ellgaard and Helenius, 2003*). Because of the continuous flow of material from the ER to the Golgi, the chaperones and redox enzymes needed for protein folding in the ER lumen undergo dynamic retrieval from the Golgi apparatus (*Gomez-Navarro and Miller, 2016*). Conversely, secretory proteins destined for secretion and integral membrane proteins intended for other cellular compartments are not retained. This separation of secreted and retained cargo proteins involves signal-mediated sorting, whereby folded proteins destined for exit from the ER have active transport or exit signals, and proteins to be retained in the ER have signals for retrieval (*Barlowe, 2003*; *Gomez-Navarro and Miller, 2016*). For membrane proteins, cytoplasmic signals can directly engage with the selective vesicle coat complexes required for transport between the ER and Golgi. For luminal proteins, this information has to be relayed by a transmembrane receptor that serves as an intermediary to the cytoplasmic coat protein complexes (*Dancourt and Barlowe, 2010*). In the archetypal KDEL-retrieval system, a seven-transmembrane receptor captures escaped ER luminal proteins carrying a C-terminal KDEL or variant tetrapeptide sequence in the mildly acidic pH of the Golgi (*Munro and Pelham, 1987*; *Semenza et al., 1990*). Signal binding to a luminal cavity in the receptor triggers a conformational change in its cytoplasmic face, exposing a lysine motif recognised by the COP I coat complex (*Bräuer et al., 2019*). Release of the signal in the neutral pH environment of the ER results in a reversal of this conformational change, burying the lysine motif, and exposing a patch of aspartate and glutamate residues on the receptor proposed to form a COPII-binding ER exit signal (*Bräuer et al., 2019*; *Newstead and Barr, 2020*). Hence, the

KDEL receptor cycles between the ER and Golgi capturing escaped ER proteins in a dynamic retrieval process (*Dean and Pelham, 1990*; *Lewis and Pelham, 1992*; *Townsley et al., 1993*; *Zagouras and Rose, 1989*). The rapid recycling of the receptor means it does not need to be stoichiometric with the ER concentration of retained proteins, only present at levels sufficient to capture escaped proteins that reach the Golgi (*Newstead and Barr, 2020*). Although ER resident proteins differ widely in concentration, remarkably, this does not pose a problem for efficient retention of the less abundant proteins. One possible explanation for this is the presence of HDEL and RDEL variants of the canonical KDEL signal with different binding affinities (*Scheel and Pelham, 1998*; *Wilson et al., 1993*). However, despite extensive mutation and structural analysis the molecular basis and functional significance of these affinity differences remains unclear (*Bräuer et al., 2019*; *Townsley et al., 1993*). Complicating this picture, in some organisms including the yeasts *Kluyveromyces lactis* and *Schizosaccharomyces pombe*, DDEL and ADEL variants are used as ER retrieval signals (*Pidoux and Armstrong, 1992*; *Semenza and Pelham, 1992*). Comparative analysis of the budding yeast *Saccharomyces cerevisiae* HDEL- and *K. lactis* DDEL-receptors implicated a luminal region including a key variant residue, D50 in the human receptor, in selectivity for DDEL (*Lewis et al., 1990*; *Semenza et al., 1990*; *Semenza and Pelham, 1992*). Mutation of D50 to cysteine in the human receptor resulted in reduced binding affinity for KDEL, RDEL, and HDEL (*Scheel and Pelham, 1998*). However, recent structure determination of the chicken receptor with a bound TAEKDEL peptide indicates this residue sits on the luminal surface of the receptor and does not make contact with any portion of the signal (*Bräuer et al., 2019*). Thus, although it is clear that the specificity of ER retrieval is encoded by the KDEL receptor, the molecular basis for the recognition of different signal variants remains unclear.

Our previous work has shown the KDEL receptor has a transporter-like architecture and undergoes pH-dependent closure around cognate retrieval signals (*Bräuer et al., 2019*; *Newstead and Barr, 2020*). However, the molecular basis for affinity differences for retrieval signal variants and any functional significance these differences may create, was not explained by that work or other previous studies. Furthermore, how signals are initially captured and selected from other sequences is not understood. To answer these related questions, we solved structures of a vertebrate KDEL receptor in complex with both HDEL and RDEL retrieval signals, and performed a combination of computational and cell biological analysis. Based on this data, we can break down the retrieval signal recognition process into a series of steps for initial capture of the free carboxyl terminus and signal proofreading, followed by full engagement with the binding cavity and finally pH-dependent closure of the receptor to expose the COPI retrieval motif.

## Results

### ER retrieval signals in mammalian cells

To understand how the KDEL-receptor differentiates between cargo proteins, we first sought to define the major signal variants used in mammalian cells. For this purpose, we exploited luminal ER proteome datasets to investigate the relative abundance of retrieval signal variants (*Itzhak et al., 2017*; *Itzhak et al., 2016*). This confirmed that KDEL, HDEL and RDEL are the major variants in mammals, and the frequency of ER resident proteins with these variants of the retrieval signal at the −4 position is approximately equal (*Figure 1a*). However, this does not reflect the abundance of the proteins carrying the signal. Strikingly, the total concentration of KDEL bearing proteins is over fivefold higher than either HDEL or RDEL (*Figure 1b*). This largely reflects a small number of highly abundant ER-resident chaperones, BIP, PDI, and calreticulin (*Figure 1—figure supplement 1a and b*). Each of these proteins is present in the 5–10 µM range, far more abundant than the dominant KDELR receptor 2 (KDELR2) species which is estimated to be 0.2–0.3 µM (*Figure 1—figure supplement 1c*). In total, the concentration of retrieval signals thus exceeds that of the receptor by at least two orders of magnitude. In good agreement with previously reported studies on the mammalian KDEL receptor (*Scheel and Pelham, 1998*; *Wilson et al., 1993*), we found that HDEL has the highest affinity for the receptor $K_D$ 0.24 µM, followed by KDEL $K_D$ 1.94 µM and RDEL $K_D$ 2.71 µM (*Figure 1c*). Previous work has suggested DDEL binds to semi-purified human KDEL receptors in membrane fractions and can function as a retrieval signal when the receptor is overexpressed at high level in COS7 cells (*Lewis and Pelham, 1992*; *Wilson et al., 1993*). However, we find that

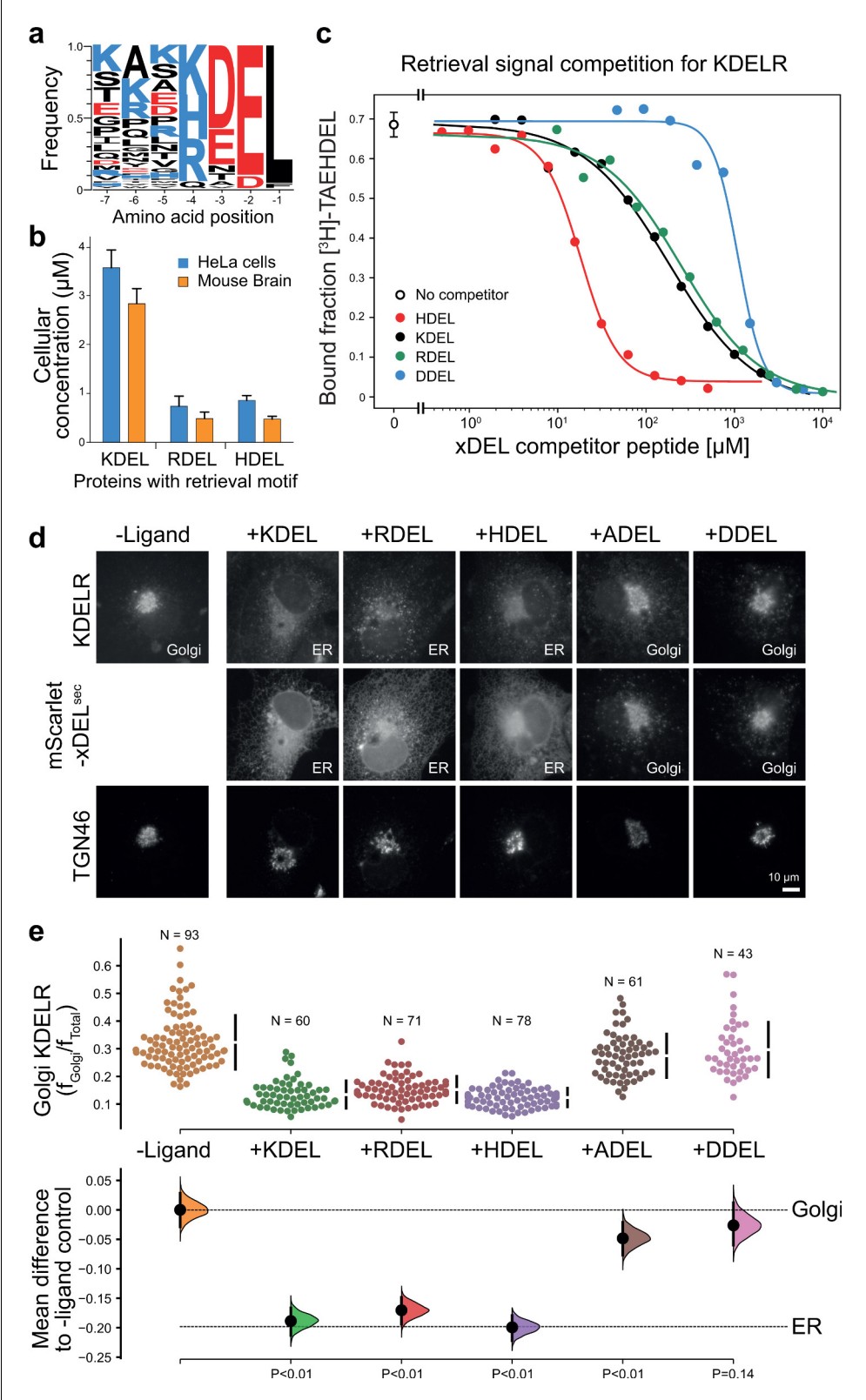

**Figure 1.** ER retrieval signal abundance and affinity are not correlated. (**a**) Sequence logos for ER resident proteins with C-terminal KDEL retrieval signals and variants thereof calculated using frequency or protein abundance (***Itzhak et al., 2017***; ***Itzhak et al., 2016***). (**b**) Combined cellular concentrations of ER resident proteins with canonical KDEL, RDEL, and HDEL retrieval sequences in HeLa cells and mouse brain. (**c**) Competition binding assays for [³H]-TAEHDEL and unlabelled TAEKDEL, TAERDEL, and TAEHDEL to the KDEL receptor. IC$_{50}$ values for the competing peptides

*Figure 1 continued on next page*

*Figure 1 continued*

were used to calculate the apparent $K_D$ with the Cheng-Prusoff equation (*Cheng and Prusoff, 1973*). (d) Endogenous KDEL receptor redistribution was measured in COS-7 cells in the absence (-ligand) or presence of K/R/H/A/DDEL (mScarlet-xDEL$^{sec}$). TGN46 was used as a Golgi marker. Scale bar is 10 µm. (e) The mean difference for K/R/H/A/DDEL comparisons against the shared no ligand control are shown as Cummings estimation plots. The individual data points for the fraction of KDEL receptor fluorescence in the Golgi are plotted on the upper axes with sample sizes and p values. The online version of this article includes the following source data and figure supplement(s) for figure 1:

**Source data 1.** Source data for the ligand-induced KDELR receptor retrieval assays in *Figure 1e*.
**Figure supplement 1.** Abundance of ER resident proteins and chaperones in human cells and mouse brain.
**Figure supplement 1—source data 1.** Source data for analysis of ER protein levels in *Figure 1—figure supplement 1a and c*.
**Figure supplement 1—source data 2.** Source data for the western blots in *Figure 1—figure supplement 1d* showing the regions taken for the figure.
**Figure supplement 1—source data 3.** Source data for analysis of ER protein levels in *Figure 1—figure supplement 1e and f*.
**Figure supplement 2.** Retrieval specificity of KDELR1 and KDELR2.
**Figure supplement 2—source data 1.** Source data for the ligand-induced KDELR receptor retrieval assays in *Figure 1—figure supplement 2*.

DDEL binds with 60-fold lower affinity than HDEL ($K_D$ 14.9 µM) (*Figure 1c*), in agreement with other data for purified KDEL receptors (*Scheel and Pelham, 1998*). Thus, the receptor binds to the HDEL sequence with one order of magnitude greater affinity than the canonical KDEL ligand present on the most abundant ER resident proteins. Despite this difference in affinities, mScarlet fusions with KDEL, RDEL or HDEL signals all triggered similar changes to the steady-state distribution of the endogenous KDEL receptor in cells, driving almost complete retrieval from the Golgi to the ER (*Figure 1d and e*). By contrast, expression of ADEL or DDEL had little effect on the Golgi-ER distribution of the receptor (*Figure 1d and e*), consistent with the far lower affinity relative to HDEL. Similar results were obtained for assays performed with exogenous KDELR1 and R2 (*Figure 1—figure supplement 2*), the major variants present in the cells used. In line with these effects on the receptor, the mScarlet-KDEL, RDEL, and HDEL ligands were retrieved to the ER, whereas ADEL and DDEL showed predominantly Golgi and punctate localisation consistent with secretion (*Figure 1d*). These latter observations explain why there are no verified examples of endogenous ER proteins using ADEL and DDEL retrieval signals in mammalian cells.

Given its higher affinity, why then is HDEL not the dominant ER retrieval signal, especially for crucial ER proteins such as BIP, PDI and calreticulin? We tested the idea that due to its higher binding affinity, increasing the concentration of HDEL bearing proteins would effectively compete for KDEL receptors in the Golgi, and prevent efficient ER retrieval of KDEL and RDEL containing proteins. To do this, we used our series of variant xDEL signals, where x at the −4 position is either K, R, H, A, or D. When expressed in cells, KDEL, RDEL, and HDEL are retained in the cell, whereas ADEL and DDEL are mostly secreted (*Figure 1—figure supplement 1d and e*). With the exception of HDEL this is broadly in line with their respective binding affinities. Despite binding to the receptor with a higher affinity (*Figure 1c*), HDEL was less efficiently retained than either KDEL or RDEL (*Figure 1—figure supplement 1d and e*). We then examined the effect of these ligands on the major ER proteins BIP and PDI as well as the less abundant chaperones ERP72 and ERP44 (*Figure 1—figure supplement 1a*). As predicted, ADEL and DDEL had little effect on ER retention, while HDEL caused secretion of all four proteins (*Figure 1—figure supplement 1f*).

These results indicate that the retrieval system is selective yet not optimised for binding affinity, and instead has evolved to ensure optimal retrieval of a broad cohort of proteins of widely differing abundance. In human cells, ADEL and DDEL do not bind to the receptor with high affinity and do not function as retrieval signals, suggesting specific recognition of the −4 position is a key determinant for binding. Previously, it has been suggested that complementary charges at receptor position 50 and the −4 position of the signal explain this specificity (*Lewis and Pelham, 1992*; *Semenza and Pelham, 1992*). However, this mechanism does not obviously explain how ADEL, with no charged residue at the −4 position, functions as a signal in some organisms. How signal selectivity is achieved was therefore a crucial question we needed to answer.

## HDEL and RDEL signals bind similarly to the canonical KDEL variant

To understand the molecular basis for the affinity differences between retrieval signal variants, we first examined the pH dependence of binding using protein stability assays. This revealed that

HDEL, KDEL and RDEL signals show similar pH dependent interaction with chicken KDELR2 (*Figure 2—figure supplement 1*). We then determined structures for chicken KDELR2 bound to HDEL and RDEL signals. These structures with TAEHDEL and TAERDEL peptides have resolutions of 2.24 and 2.31 Å, respectively (*Figure 2a–c* and *Supplementary file 1*). In both instances the overall structure of the receptor is similar to our previous complex with the TAEKDEL peptide (*Figure 2d*), with a root mean square deviation (R.M.S.D.) of 0.223 and 0.153 Å over 200 $C_\alpha$ atoms for the HDEL and RDEL structures, respectively. Both HDEL and RDEL peptides are bound in a vertical orientation with respect to the membrane, with the side chains clearly resolved in the electron density map (*Figure 2—figure supplement 2a and b*). Both the HDEL and RDEL peptides interact with the receptor through the same salt bridge interactions seen for the KDEL peptide (*Figure 2b–d*). Superimposing the three peptides reveals little movement of the peptide at the −1 and −2 positions when bound to the receptor (*Figure 2e*). For RDEL, we observe slight movement of the backbone $C_\alpha$ atom of the peptide to accommodate the larger arginine side chain, resulting in a minor repositioning of the glutamate at the −3 position in the receptor. Nonetheless, the position of the positive charge at the −4

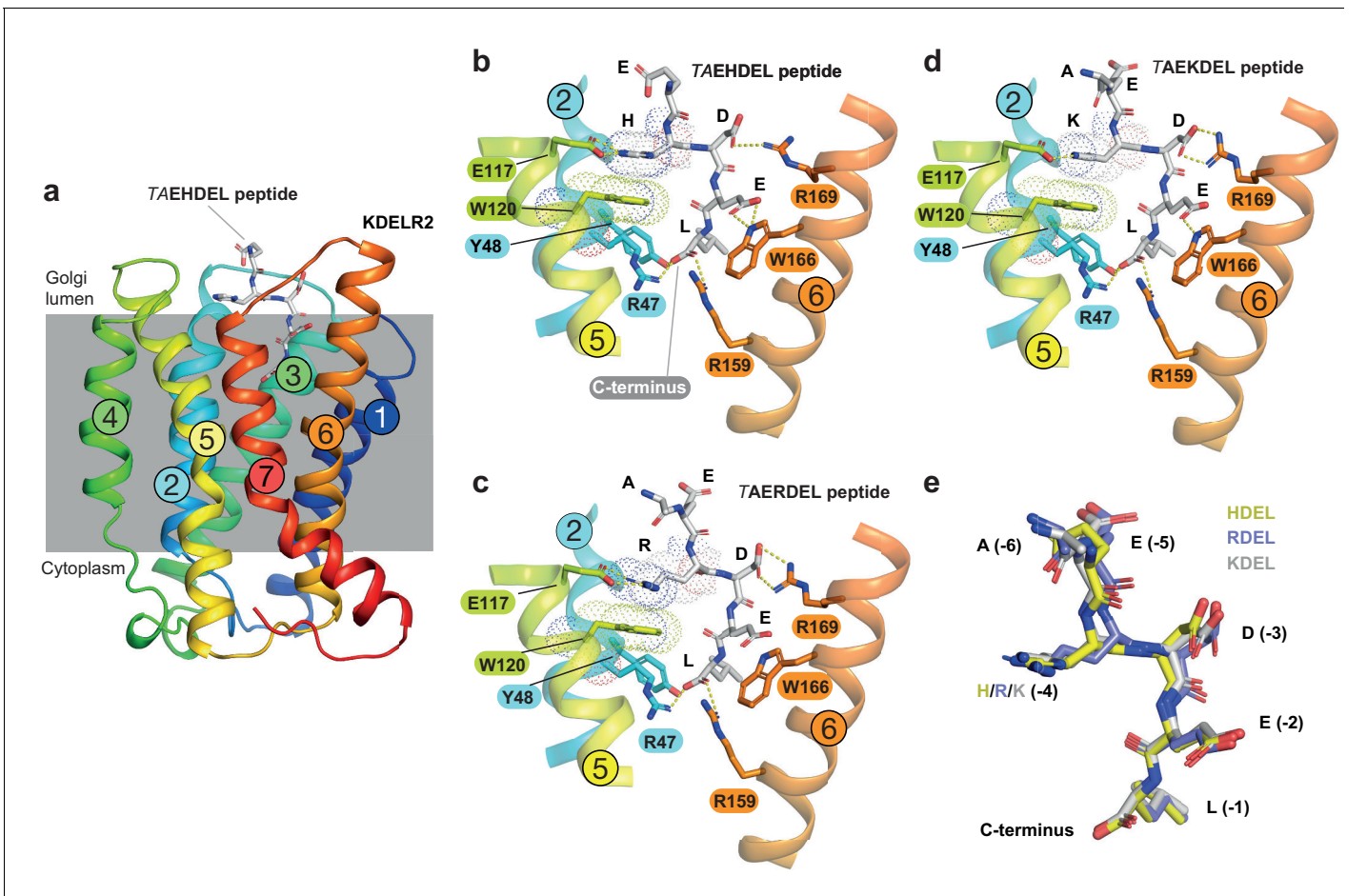

**Figure 2.** Structures of the KDEL receptor bound to HDEL and RDEL retrieval signals. (a) Crystal structure of chicken KDELR2 viewed from the side with the transmembrane helices numbered and coloured from N-terminus (blue) to C-terminus (red). The predicted membrane-embedded region of the receptor is indicated by a grey shaded box, with labels at the luminal and cytoplasmic faces. The TAEHDEL peptide is shown in stick format, coloured grey. (b) Close up views of bound TAEHDEL (this study), (c) TAERDEL (this study), and (d) TAEKDEL (PDB:6I6H) peptides bound to the receptor are shown with contributing side chains labelled. Hydrogen bonds are indicated as dashed lines. The molecular orbitals of W120 and the −4 histidine on the peptide are shown as a dotted surface. (e) Superposition of the HDEL, RDEL, and KDEL peptides reveals near identical binding position within the receptor. Retrieval signal side chains are numbered counting down from the C-terminus.

The online version of this article includes the following figure supplement(s) for figure 2:

**Figure supplement 1.** Analysis of pH-dependent interaction of HDEL, KDEL, and RDEL signals with KDELR2.

**Figure supplement 2.** Polder difference density electron density maps for HDEL and RDEL peptides.

position on all three peptides is identical relative to E117 and W120 within the receptor, supporting the view that a salt bridge is formed with E117 on TM5. D50 previously proposed to be important for recognition of the −4 position is >5 Å away, outside the region depicted in the figures, indicating it is unlikely to form a salt bridge and directly contribute to binding of the retrieval signal. Some studies have suggested the core tetrapeptide retrieval motif should be extended to include the −5 and −6 positions (*Alanen et al., 2011*). However, these positions are not conserved in retained ER luminal proteins (*Figure 1a*). In our structures, the glutamate at the −5 position sits close to S54, but would not obviously increase the binding affinity, whereas no contacts are made to the −6 position. In all cases, the free carboxy terminus and leucine residue at the −1 position form interactions to R47 and Y48 on TM2, as well as R159 and Y162 on TM6. The glutamate at position −2 forms a further salt bridge interaction to R5 on TM1 and a hydrogen bond to W166 on TM6, whereas the aspartate at −3 forms a salt bridge with R169, also on TM6. For the histidine side chain at the −4 position of HDEL, the imidazole group is predicted to form a π-π stacking interaction with W120 (*Figure 2b*). For RDEL, the arginine side chain sits in the same position as the amine group of KDEL and could thus interact with W120 via a cation-π interaction and E117 via a classical salt bridge interaction (*Figure 2c*). We therefore conclude that both E117 and W120 play a role in retrieval signal binding, and the only major difference between HDEL, RDEL, and KDEL is the precise nature of the interaction with W120 indicating that this may be a critical residue to explain the differences in affinity.

## Probing the importance of E117 and W120 for signal binding

To directly test the requirement for E117 and W120 in signal recognition, ligand binding assays using specific peptides and recombinant chicken wild type, E117 or W120 mutant KDELR2 were performed (*Bräuer et al., 2019*). All proteins had similar thermal stability indicating they were correctly folded. For the wild-type receptor at pH 5.4, $K_D$ for KDEL and HDEL peptides were 1.9 ± 0.46 µM and 0.26 ± 0.04 µM, respectively (*Figure 3a and b*). Conservative substitution of E117 with aspartate resulted in a slight reduction in binding for both KDEL and HDEL, with $K_D$ values of 2.1 ± 0.33 µM and 0.52 ± 0.02 µM, respectively (*Figure 3a and b*). Substitution of E117 with alanine had a greater effect on KDEL binding, $K_D$ ~9.3 ± 1.0 µM, compared to HDEL, $K_D$ 0.52 ± 0.02 µM (*Figure 3a and b*), suggesting that the salt bridge to E117 plays a greater role for KDEL than HDEL.

We next examined the contribution of W120 to signal recognition. Tryptophan side chains have long been recognised as important contributors in protein ligand interactions, as they are capable of interacting with ligands via both aromatic and charged forces (*Dougherty, 1996*; *Liao et al., 2013*; *Okada et al., 2001*). Our structures show that histidine, arginine or lysine side chains at the −4 position of the retrieval signal can in principle interact favourably with W120 via cation-π interactions. We reasoned that, given the additional π-π stacking observed with the imidazole group in the crystal structure, this interaction might explain the increased affinity observed for the HDEL signal variant. Accordingly, mutation of W120 to alanine resulted in loss of binding to the KDEL peptide and it was not possible to calculate a $K_D$ (*Figure 3a*). For the HDEL peptide, binding was reduced to 20% confirming that W120 plays an important role in mediating receptor-peptide interactions (*Figure 3b*). Consistent with the hypothesis that the histidine of HDEL undergoes π-π stacking interactions with W120, conserved substitution to phenylalanine supported 50% HDEL binding with $K_D$ 5.5 ± 0.57 µM, whereas no interaction was observed with the KDEL peptide (*Figure 3a and b*). Thus, W120 plays a crucial role in binding of both KDEL and HDEL and may explain the higher affinity of the receptor for HDEL signals. In contrast, E117 is less important than initially appeared and it is unclear why it is a conserved feature of the binding site.

To analyse whether the properties measured using purified components in vitro reflect the behaviour of the KDEL receptor and retrieval system in vivo, we analysed the ability of these same variants in the human KDEL receptor to differentiate between human retrieval signal sequences in a cellular ER retrieval assay. All the receptor mutants tested reached the Golgi apparatus supporting the view they are able to fold and exit the ER (*Figure 3c*, -Ligand, and *Figure 3—figure supplement 1a*). The WT receptor showed robust retrieval to the ER in response to model cargo proteins bearing KDEL, RDEL, or HDEL sequences (*Figure 3c* and *Figure 3—figure supplement 1a–d*). Receptors with conservative (E117Q and E117D) or non-conservative (E117A) substitutions at E117 were efficiently retrieved to the ER with KDEL, RDEL, or HDEL signal variants (*Figure 3c* and *Figure 3—figure supplement 1b–d*). In contrast, receptors with mutations at W120A and W120F did not respond to KDEL and RDEL signals and showed greatly reduced response to HDEL (*Figure 3c* and *Figure 3—*

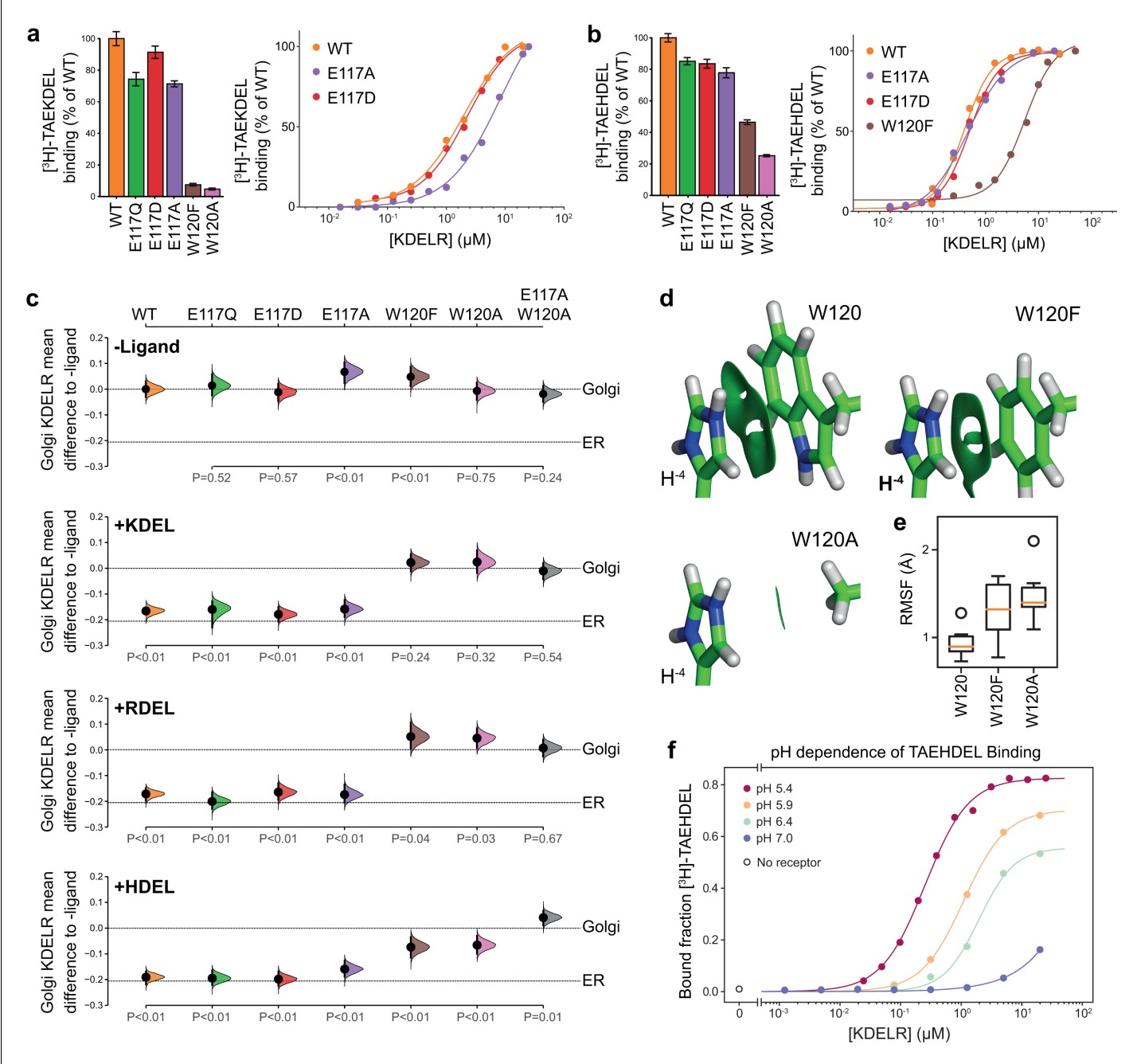

**Figure 3.** Roles of KDEL receptor E117 and W120 in retrieval signal binding and function in cells. (a) Normalised binding of [³H]-TAEKDEL and (b) [³H]-TAEHDEL signals to purified WT and the indicated E117 and W120 mutant variants of chicken KDELR2. Bar graphs show mean binding ± SEM (n = 3). Line graphs show titration binding assays. (c) The fraction of WT, E117, and W120 mutant KDEL receptor localised to the Golgi in COS-7 cells was measured before (no ligand) and after challenge with different retrieval signals (K/R/HDEL) as indicated. Effect sizes are shown as the mean difference for K/R/HDEL comparisons against the shared -ligand control with sample sizes and p-values. Also see *Figure 3—figure supplements 1—source data 1* files. (d) The π-π interactions between W120 and the histidine were visualised using reduced density gradient analysis. The wild-type W120 exhibit stronger π-π interactions compared with W120F, while W120A shows no π-π interactions. (e) When W120 is changed to phenylalanine, the protonated histidine has a higher root mean squared fluctuation (RMSF) in the binding pocket, which is further increased for the W120A substitution. (f) Binding of [³H]-TAEHDEL to the KDEL receptor was measured at pH 5.4–7.0 and is plotted as a function of receptor concentration.

The online version of this article includes the following source data and figure supplement(s) for figure 3:

**Figure supplement 1.** Effect of KDEL receptor E117 and W120 mutants on retrieval signal function in cells.

**Figure supplement 1—source data 1.** Source data for the ligand-induced KDELR receptor retrieval assays in *Figure 3* and *Figure 3—figure supplement 1*.

*figure supplement 1b–d*). The residual response to HDEL was abrogated in a double E117A/W120A mutant receptor (*Figure 3c* and *Figure 3—figure supplement 1b–d*). This in vivo behaviour is in good agreement with the changes to affinity measured using in vitro binding assays (*Figure 3a and b*), and supports the view that W120 is of greater importance for ligand binding and ER retrieval.

To provide further support for this conclusion, we investigated the free energy of interaction between the histidine side chain of the retrieval signal and W120 of the receptor. Protonation of the HDEL histidine is a crucial consideration since retrieval signal binding to the receptor occurs at mildly acidic pH in the Golgi. We therefore asked if the protonation state of the histidine is important for binding affinity. Molecular mechanics-based alchemical transformation was used to compute the free energy difference of changing the lysine in KDEL to different protonation states of the histidine in HDEL. The binding free energy of HDEL is $-1.8 \pm 1.4$ kcal.mol$^{-1}$ stronger than the KDEL signal (*Supplementary file 2*), which is in good agreement with the expected $-1.3$ kcal.mol$^{-1}$ free energy difference derived from measured $K_D$ values for KDEL and HDEL. The preference for HDEL of $-1.9 \pm 0.2$ kcal.mol$^{-1}$ is mainly attributed to the protonated histidine, pKa calculated to be $8.9 \pm 0.5$, which makes favourable cation-$\pi$ interactions with W120 (*Supplementary file 2*, HIP). In agreement with the experimental data (*Figure 3b and c*), the W120F mutation, which is anticipated to preserve the cation-$\pi$ interactions, shifts the pKa to $7.6 \pm 0.3$ and reduced but did not abolish the preference for HDEL to $-0.7 \pm 1.6$ kcal.mol$^{-1}$, notwithstanding the large error on this calculation. Furthermore, the W120A mutation which eliminates the cation-$\pi$ interactions, gave a side chain pKa of $6.5 \pm 0.1$ and greatly reduced the preference for HDEL to $-0.3 \pm 0.9$ kcal.mol$^{-1}$.

To quantify the strength of the $\pi$-$\pi$ and cation-$\pi$ interactions between W120 variants and the histidine, we decomposed the interactions using symmetry-adapted perturbation theory from quantum mechanics. Although both W120 and W120F form $\pi$-$\pi$ and cation-$\pi$ interactions with protonated histidine, W120F exhibits ~1.5 kcal.mol$^{-1}$ weaker $\pi$-$\pi$ interactions and ~0.5 kcal.mol$^{-1}$ weaker cation-$\pi$ interactions with the histidine (*Figure 3d* and *Supplementary file 3*). The consequence of these changes is that for W120F higher root mean squared fluctuations are seen (*Figure 3e*), indicative of less rigid binding. These fluctuations are further increased for W120A (*Figure 3e*), consistent with its greater effect on signal binding. These results support the hypothesis that the $\pi$-$\pi$ interactions between the protonated histidine sidechain and W120 explain the higher affinity observed for HDEL signals. Further support for this interpretation comes from in vitro analysis of the pH-dependence of HDEL binding. At pH 6.4, HDEL shows ~ 60% maximal binding to the receptor (*Figure 3f*), compared to <20% seen at the same pH for KDEL (*Bräuer et al., 2019*). The level of HDEL binding seen at pH 7 would saturate the KDELR receptor in the ER if the most abundant luminal proteins such as BIP carried this signal variant. Our observation that W120 is also necessary for recognition of KDEL indicates that cation-$\pi$ interactions to W120, rather than a salt bridge to E117, is the crucial determinant for recognition of the $-4$ position.

## E117 plays a role in KDEL receptor selectivity

This mode of signal binding involving W120 is different than previously proposed, where charge complementarity between D50 in the receptor and the $-4$ position of the signal was thought to be a key determinant of specificity in ER retrieval (*Lewis and Pelham, 1992*; *Scheel and Pelham, 1998*; *Semenza and Pelham, 1992*). However, as our crystal structures show, D50 is outside the immediate binding region for all retrieval signal variants and therefore unlikely to directly contribute to binding. Thus, the precise roles of D50 and E117 remain poorly defined. In this regard, the behaviour of ADEL signals is noteworthy due to the simple methyl side chain. Comparison of different retrieval signals shows that ADEL does not activate the wild-type human KDEL receptor (*Figure 1d and e*). The simplest explanation for this finding is that the $-4$ position is crucial for high-affinity binding of retrieval signals to the human receptor. Nonetheless, this simple view is unlikely to be correct. First, the KDEL, RDEL, and HDEL-bound receptor structures do not support the view that recognition of the $-4$ position requires D50, and instead provide an alternative possibility where E117 fulfils this role. Second, our biochemical and functional data show that E117 does not contribute greatly to signal binding affinity or retrieval in cells (*Figure 3a–c*). Therefore, rather than selecting for the sequence, E117 may be more important to select against unwanted signal variants, perhaps on the basis of their net charge. To test this idea, we examined the response of E117A mutant receptors to variant ADEL and DDEL signals. Remarkably, the E117A mutant receptor relocated to the ER in

response to both KDEL and ADEL, but not DDEL signals (*Figure 4a and b*). In *S. pombe* and *K. lactis*, organisms where ADEL and DDEL are used for ER retrieval, the E117 position of the receptor is either an asparagine or a glutamine residue, and we therefore tested E117N and E117Q mutants. Similar to the results with E117A, E117Q, and E117N receptors move to the ER in response to KDEL or ADEL signals, yet interestingly still failed to respond to DDEL (*Figure 4a and b*). Ligand expression was in a similar range in all instances (*Figure 4—figure supplement 1*), and in the absence of ligand all three mutant receptors localised to the Golgi with a low ER background indicating normal folding and ER exit (*Figure 4a*).

Thus, E117 is important for determining which signals are rejected by the wild type human receptor based on the −4 position of the signal, but does not appear to play a major role in binding affinity. ADEL must bind to the E117A mutant receptors via the 'DEL' tri-carboxylate portion of the retrieval signal, suggesting this region may be the major contributor to binding affinity for all signal variants. For HDEL, the protonated histidine side chain makes additional π-π interactions with W120 to bind with higher affinity. Importantly, the lack of response to DDEL shows that signal selection and recognition must involve additional features in the *S. pombe* and *K. lactis* receptor, and we investigated this question further.

## A charge screening mechanism for signal differentiation by the KDEL receptor

To identify additional features that might play a role in signal selection, we performed a comparison of the receptors and most abundant cognate ligands of the HSPA5/BIP family of ER resident proteins in different species. Although most regions of the receptor are highly conserved, as noted previously (*Semenza and Pelham, 1992*), sequence alignment reveals two regions where there is

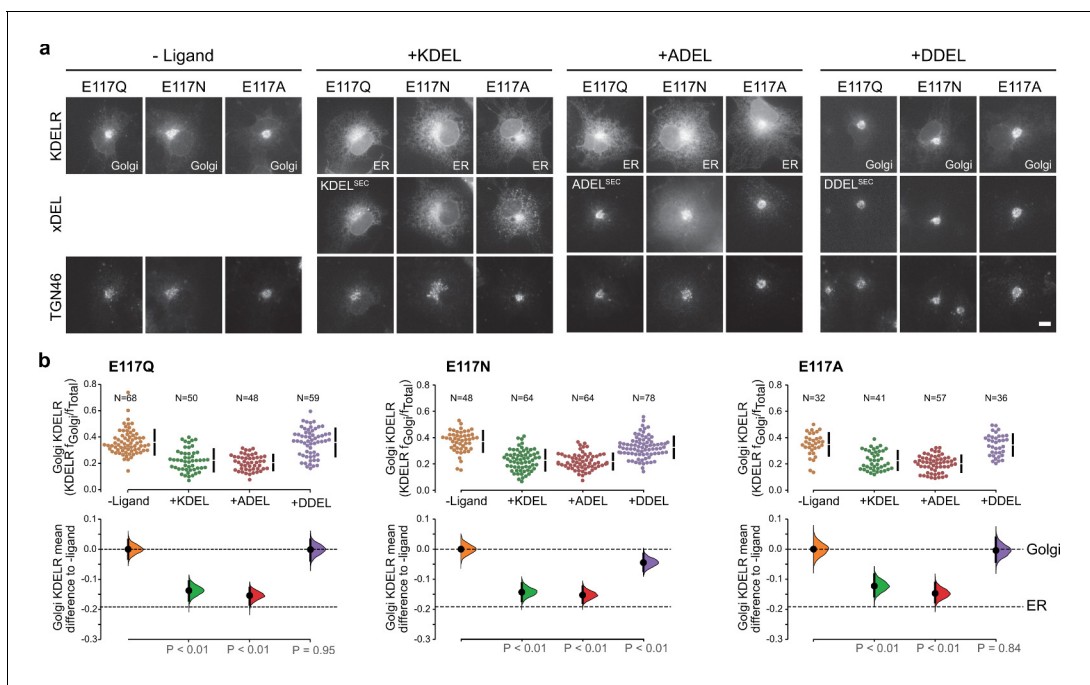

**Figure 4.** KDEL receptor E117 mutants show reduced selectivity for retrieval signals. (a) E117Q, E117N, or E117A mutant KDEL receptors were tested for K/A/DDEL-induced redistribution from Golgi to ER in COS-7 cells. KDEL receptor distribution was followed in the absence (-ligand) or presence of K/A/DDEL^sec. TGN46 was used as a Golgi marker. Scale bar is 10 μm. (b) The fraction of E117Q, E117N or E117A mutant KDEL receptor localised to the Golgi was measured before (no ligand) and after challenge with different retrieval signals (K/A/DDEL). Effect sizes are shown as the mean difference for K/A/DDEL comparisons against the shared -ligand control with sample sizes and p values. .

The online version of this article includes the following source data and figure supplement(s) for figure 4:

**Source data 1.** Source data for the ligand-induced KDELR receptor retrieval assays in *Figure 4*.

**Figure supplement 1.** Effect of ligand levels on the response of KDEL receptor E117 mutants to KDEL, ADEL, and DDEL signals.

**Figure supplement 1—source data 1.** Source data for the ligand-induced KDELR receptor retrieval assays in *Figure 4—figure supplement 1*.

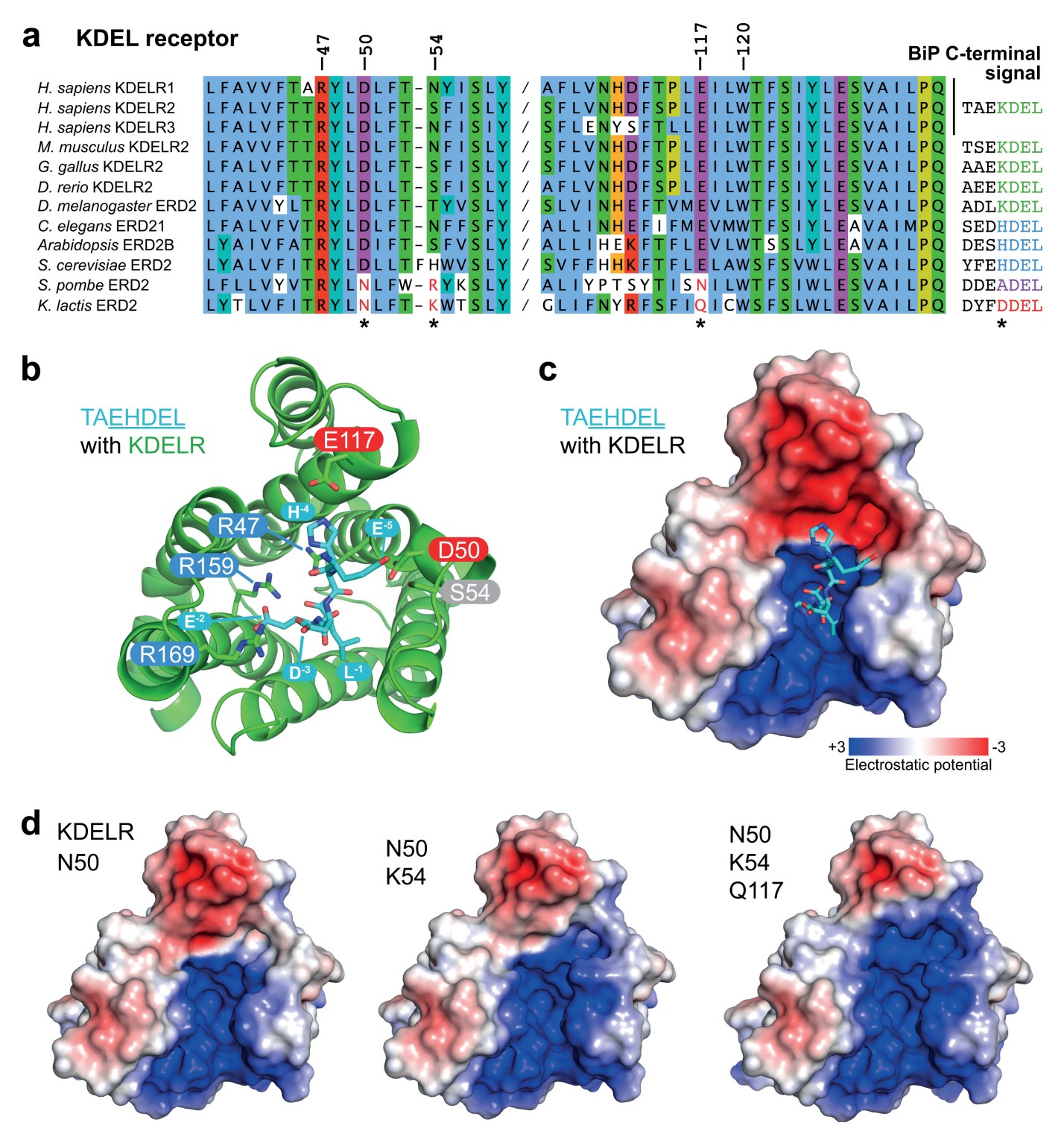

**Figure 5.** Charge distribution across the luminal entrance to the KDEL receptor binding pocket. (**a**) KDEL receptor sequence alignment showing two regions centred around amino acid D50 and W120 of the human proteins. Cognate retrieval signal variants are shown to the right of the alignment. (**b**) The structure of the KDEL receptor with bound TAEHDEL highlighting key residues involved in ligand binding and variant residues D50, N54, and E117. (**c**) The charged surface for the WT KDEL receptor and (**d**) N50, N50/K54 and N50/K54/Q117 mutants is shown.

The online version of this article includes the following source data and figure supplement(s) for figure 5:

**Figure supplement 1.** Comparison of human and yeast ER retrieval signals.

*Figure 5 continued on next page*

*Figure 5 continued*

**Figure supplement 1—source data 1.** Source data for the ligand-induced KDELR receptor retrieval assays in *Figure 5—figure supplement 1*.

covariation that may be related to the cognate tetrapeptide retrieval signal (*Figure 5a*). In receptors recognising ADEL and DDEL, D50 is changed for asparagine, E117 for glutamine or asparagine, and position 54 is a positively charged arginine or lysine rather than a polar side chain (*Figure 5a*). To understand the consequences of these changes we examined their positions relative to the bound TAEHDEL signal (*Figure 5b*). This reveals that E117 and S54 sit close to the −4 histidine and −5 glutamate, respectively and D50 is over 5 Å away from any residue in the signal in the final bound state (*Figure 5b*). Analysis of the charge distribution across the surface of the receptor shows a negatively charged feature above the positively charged binding cavity occupied by the DEL portion of the signal, with the −4 residue sited at the boundary to these two regions (*Figure 5c*). Strikingly, progressive introduction of changes in the human receptor to mimic the *K. lactis* receptor, D50N S/N54K E117Q erodes the negatively charged luminal feature (*Figure 5d*).

One simple explanation for this feature is that it extends the binding site to impart specificity for the region upstream of the core KDEL signal. However, analysis of different classes of ER luminal proteins from yeast and animal cells does not provide strong support for this possibility. The upstream sequences of many abundant ER proteins including human and yeast HSPA5/BIP homologues are acidic in nature, and not basic (*Figure 5a* and *Figure 5—figure supplement 1a*), making any interaction unfavourable. For the human signal, the −4 position is crucial and mutation to A or D abolishes ER retrieval of the receptor (*Figure 5—figure supplement 1b* and *Figure 5—figure supplement 1d*). Conversely, *S. pombe* and *K. lactis* BIP ADEL and DDEL signals become functional with the human receptor if the −4 position is changed to lysine confirming this is the critical residue, independent of upstream sequences (*Figure 5—figure supplement 1c and d*). In *K. lactis* BIP the −5 position is a bulky aromatic residue rather than a charged residue. Previous work has suggested that the budding yeast FEHDEL signal with a bulky aromatic residue at the −6 position does not function in mammalian cells (*Wilson et al., 1993*), however consistent with our other data we find this HDEL variant is also functional (*Figure 5—figure supplement 1c*). Extending this analysis to human FKBP family proteins with even more diverse upstream sequences reveals no obvious pattern of conservation other than the canonical C-terminal HDEL or HEEL retrieval signal (*Figure 5—figure supplement 1e*).

To directly test the role of the charged luminal surface in signal selection, we made a series of mutants introducing the changes seen in *K. lactis* and *S. pombe* into the human receptor and tested these against KDEL, ADEL, and DDEL signals. A single D50N mutation abolished the response to all signal variants and the receptor remained in the Golgi (*Figure 6a and b*). Thus, like E117, D50 is not the sole determinant of signal selectivity. Similarly, N54K or N54R reduced the response to KDEL but did not result in ADEL or DDEL recognition (*Figure 6a and b*; *Figure 6—figure supplement 1a and b*). D50N N54K and D50N N54R double mutants showed a loss of specificity and gave a response to KDEL, ADEL, and DDEL signals, showing that it is possible to uncouple binding from selectivity at the −4 position (*Figure 6b* and *Figure 6—figure supplement 1b*). We then combined D50N or N54K with E117Q mutations. These *K. lactis* like double mutant receptors showed switched specificity towards ADEL and DDEL with only a residual response to KDEL (*Figure 6a and b*). Combination of D50N N54K and E117Q improved the response to ADEL and DDEL and further reduced that towards KDEL (*Figure 6a and b*). Comparable results were obtained with a *S. pombe* like D50N N54R E117N triple mutant receptor (*Figure 6—figure supplement 2a and b*). Both these altered specificity receptors responded to the cognate ADEL or DDEL variant of BIP for that organism, a response that was abolished solely by mutation of the −4 position of the signal (*Figure 6—figure supplement 2a and b*).

These results indicate that the −4 position of the signal is read out during initial signal binding and is important for exclusion of unwanted signals, but is less important for binding affinity. We therefore tested whether the mode of ADEL and DDEL binding to the switched specificity receptors still involves W120. A D50N N54K E117Q W120A *K. lactis* like mutant receptor does not relocate from the Golgi to the ER with KDEL and ADEL signals and shows only a small response to the DDEL signal (*Figure 6a and b*). Similar results were obtained with a *S. pombe* like D50N N54R E117N

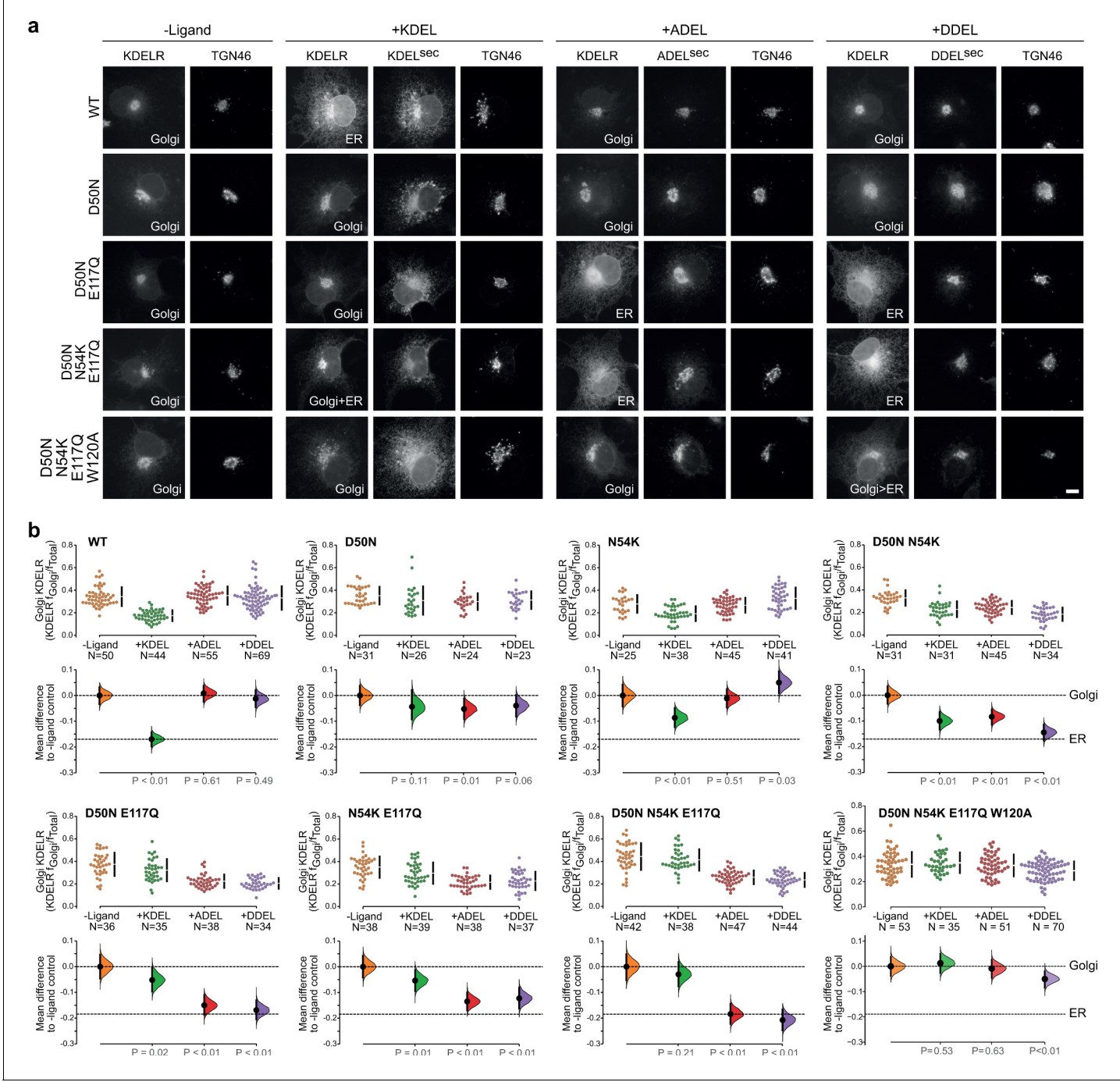

**Figure 6.** Re-engineering the selectivity of the human KDEL receptor for ADEL and DDEL signals. (**a**) WT and a series of 'K. lactis'-like mutant KDEL receptors were tested for K/A/DDEL-induced redistribution from Golgi to ER in COS-7 cells. KDEL receptor distribution was followed in the absence (-ligand) or presence of K/A/DDEL^sec. TGN46 was used as a Golgi marker. Scale bar is 10 μm. (**b**) The fraction of WT and mutant KDEL receptor localised to the Golgi was measured before (no ligand) after challenge with different retrieval signals (K/A/DDEL). Effect sizes are shown as the mean difference for K/A/DDEL comparisons against the shared -ligand control with sample sizes and p values.

The online version of this article includes the following source data and figure supplement(s) for figure 6:

**Source data 1.** Source data for the ligand-induced KDELR receptor retrieval assays in *Figure 6*.
**Figure supplement 1.** Extended analysis of human KDEL receptor selectivity.
**Figure supplement 1—source data 1.** Source data for the ligand-induced KDELR receptor retrieval assays in *Figure 6—figure supplement 1*.
**Figure supplement 2.** Retrieval specificity of 'K. lactis' and 'S. pombe' triple mutant KDEL receptors.
**Figure supplement 2—source data 1.** Source data for the ligand-induced KDELR receptor retrieval assays in *Figure 6—figure supplement 2*.

W120A mutant receptor, albeit with some remaining response to DDEL (*Figure 6—figure supplement 1b*). Together, these findings suggest a common mode of binding for all retrieval signal variants through conserved residues. Specificity for the −4 position is largely achieved through a proofreading mechanism involving gatekeeper residues, D50 and E117, as the signal enters the ligand-binding cavity. Additionally, S/N54 contributes to the exclusion of unwanted signal variants. An E117A substitution partially uncouples this mechanism and allows ADEL binding, whereas both D50 and E117 residues have to be changed to allow DDEL binding. Bringing together all our observations to this point, we conclude that the luminal surface of the receptor plays a crucial role in signal selectivity prior to adoption of the final activated state, perhaps by determining the rate of signal association from solution.

## Initial retrieval signal capture by the free carboxyl terminus

To explore the initial interaction of retrieval signals with the KDEL receptor, we simulated an all-atom model of the KDEL signal with a free C-terminal carboxylate engaging with the receptor (*Video 1*). This simulation shows that the signal initially encounters the receptor through a salt bridge interaction from its C-terminal carboxyl group with R169 on TM6 of the receptor (*Figure 7a, i.*). The C-terminal carboxyl group then moves to engage R5 (*Figure 7a,ii.*), shortly followed by interaction of the glutamate −2 with R169 (*Figure 7a,iii.*). Finally, the C-terminus engages with R47 on TM4 enabling aspartate −3 to interact with R169 (*Figure 7a, iv.*). Thus, the carboxy-terminus of the retrieval signal sequentially engages R169, R5 and finally R47 (*Figure 7c*). Movement of the lysine at the −4 position towards E117 is concomitant with the final engagement of the carboxyl-terminus of the signal by R47, whereas D50 does not come in close proximity to the KDEL signal and there is only a transient interaction of S54 with the −5 position (*Figure 7d*).

We therefore propose a carboxyl-handover model for signal capture mediated by the ladder of arginine residues in the binding pocket (*Figure 7b*). As the carboxyl-terminus progresses further into the receptor-binding site, the carboxylate groups at positions −2 and −3 engage their respective positions in the D- and E-sites, respectively. Only the final stage of the binding, where the receptor closes around the signal locking it in place is pH dependent, all other stages are predicted to be freely and rapidly reversible. Because many proteins have a free C-terminal carboxylate, this highlights the importance of an initial proofreading stage where non-cognate signals are rejected, as we have already argued, due to their net charge.

To test these ideas, we investigated the importance of the retrieval signal C-terminus and R169 in the receptor using in vitro binding assays and functional experiments in cells. First, we synthesised C-terminally amidated HDEL and KDEL peptides and assayed their ability to bind to wild-type receptors (*Figure 7e*). Blocking the C-terminal carboxylate in this way completely abolished binding to KDEL and reduced the affinity for the HDEL peptide by two orders of magnitude from 19 ± 1.3 µM to 1.7 ± 0.1 mM. For HDEL, this residual affinity suggests the peptide still enters and exits the binding pocket, but fails to trigger the final pH dependent capture. Next, we performed binding assays with R169 variant receptors. Comparable results to the C-amidated peptide binding assays were obtained with R169A, which showed no binding to KDEL and greatly reduced binding to HDEL ligands (*Figure 7f*). Conservative substitution to R169K greatly reduced binding of both HDEL and KDEL in line with predictions (*Figure 7f*). Finally, we tested the R169 variants in ER retrieval assays. R169A mutant receptors showed no response to KDEL and only ~10% response to HDEL signals (*Figure 7g*, *Figure 7—figure supplement 1a and b*). By contrast, the conservative substitution R169K showed an attenuated response to both signals, in agreement with the simulation and

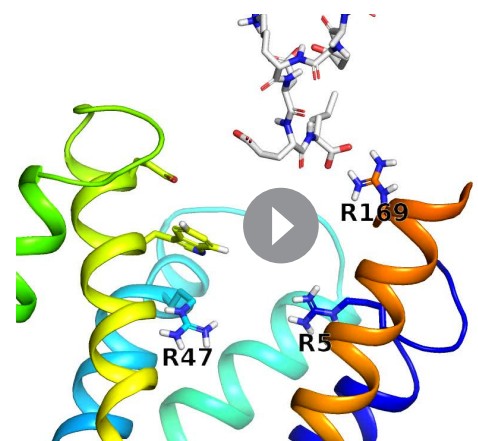

**Video 1.** Stepwise engagement of the KDEL signal with the KDEL receptor. Molecular dynamics simulation of TAEKDEL binding to the KDEL receptor simulated using molecular dynamics.
https://elifesciences.org/articles/68380#video1

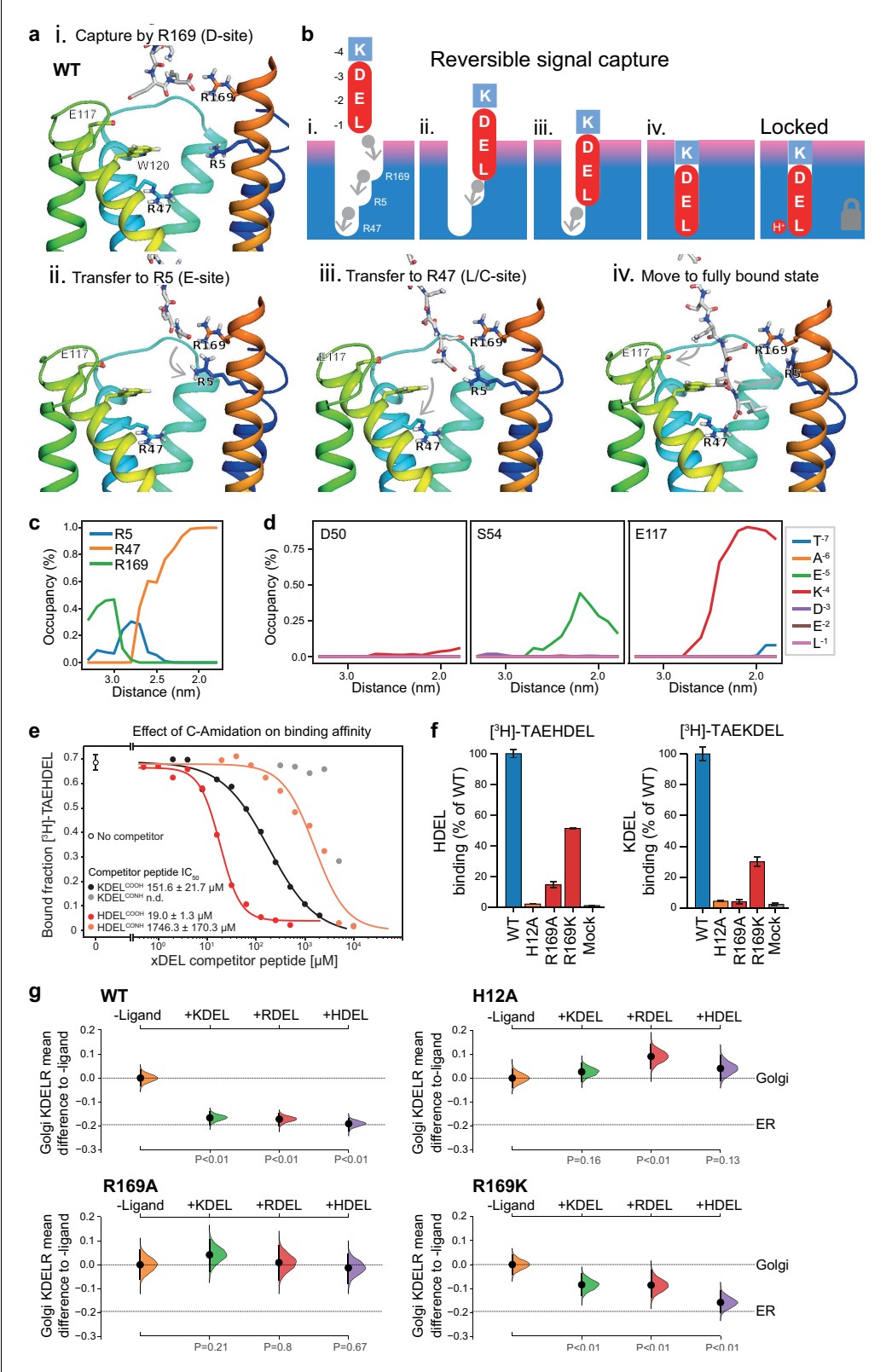

**Figure 7.** Mechanism for initial retrieval signal capture by the KDEL receptor. (**a**) Images depicting the key stages (i.-iv.) of TAEKDEL binding to the wild-type (WT) KDEL receptor simulated using molecular dynamics. Initial engagement of the C-terminus to R169 (i) is followed by transfer to R5 (ii), shortly followed by interaction of E −2 with R169 (iii). Finally, R47 engages the C-terminus allowing D −3 to interact with R169 (iv). See also *Video 1*. (**b**) A carton model depicting the key stages of retrieval signal binding and final pH-dependent locked state. (**c**) Occupancy of the hydrogen bonds

*Figure 7 continued on next page*

*Figure 7 continued*

between the C-terminus of the KDEL retrieval signal and R5, R47, and R169 is plotted as a function of signal position within the binding pocket. (d) The occupancy of potential hydrogen bonds between the different positions of the KDEL retrieval signal and D50, S54, and E117 is plotted as a function of signal position within the binding pocket. (e) Competition binding assays for [$^3$H]-TAEHDEL and unlabelled TAEKDEL and TAEHDEL with a free (COOH) or amidated (CONH) C-terminus to chicken KDELR2 showing $IC_{50}$ values for the competing peptides. (f) Normalised binding of [$^3$H]-TAEHDEL and [$^3$H]-TAEKDEL signals to the purified WT H12A, R169A, or R169K mutant chicken KDELR2. A mock binding control with no receptor indicates the background signal. (g) Distribution of WT, H12A, R169A, and R169K KDEL receptors was measured in COS-7 cells in the absence (-ligand) or presence of K/R/HDEL$^{sec}$. The mean differences for K/R/HDEL comparisons against the shared no ligand control are shown with sample sizes and p values. See also *Figure 7—figure supplement 1* with accompanying source data.

The online version of this article includes the following source data and figure supplement(s) for figure 7:

**Figure supplement 1.** R169 plays a crucial role in signal recognition.

**Figure supplement 1—source data 1.** Source data for the ligand-induced KDELR receptor retrieval assays in *Figure 7* and *Figure 7—figure supplement 1*.

reduced binding affinity (*Figure 7g*, *Figure 7—figure supplement 1a and b*). We therefore conclude that the interaction of receptor R169 with the C-terminal carboxylate of the retrieval signal plays an important role in initial signal capture.

## Discussion

### A mechanism for initial signal capture and proofreading by the KDEL receptor

Canonical ER retrieval signals can be broken down into two components: the −4 position, which enables the receptor to distinguish between different populations of ER proteins, and a tri-carboxylate moiety formed by the −3 aspartate, −2 glutamate and −1 C-terminal carboxylate. We propose a relay handover mechanism for capture of this signal by the KDEL receptor wherein a ladder of three arginine residues in the receptor pairs with the three-carboxyl groups of the signal (*Figure 8*). During cargo capture, the receptor engages the retrieval signal in a stepwise process, with the C-terminal carboxyl group of the cargo protein moving between these three interaction sites. At neutral pH, C-terminal sequences will rapidly sample the binding site, a process that we imagine will occur in both the ER and Golgi apparatus. This is accompanied by a proofreading process depending on the net charge on the signal and the gatekeeper residues D50, S54, and E117 at the entrance of the ligand-binding pocket. Although D50 and S54 do not sit close to the signal in the final bound state, they are at a similar height to E117. Our structures presumably represent the end stage of the binding process with the lowest energy state, and we have used MD to probe intermediates in the binding process. This approach combined with our functional analysis suggests that D50 and S54, or the equivalent residues, make transient interactions to the retrieval signals and provide an entry point for the signal. If so, the mechanism could be equivalent to the insertion of a key, where the retrieval signal initially binds to D50 and S54, and the relay made of three negatively charged groups on the retrieval signal and three positively charged arginine residues in the receptor drives the key into the lock. The final step, locking the C-terminus of the signal in place depends on protonation of the receptor in the Golgi as we have explained previously (*Bräuer et al., 2019*). Finally, the −4 position would rotate towards E117 to adopt the most stable low energy binding pose. This mechanism explains why the retrieval signal must be located at the C-terminus of the cargo protein, and the defined requirement for either glutamate or aspartate residues at the −2 and −3 positions due to their carboxyl group containing side chains. Variation at the −4 position would not directly alter the initial capture and relay mechanism, possibly explaining why it is the key determinant for signal selectivity.

The structures we have obtained for the KDELR2 with bound HDEL, RDEL or KDEL signals reveal that the side chains at the −4 position form a salt bridge interaction with E117 but, crucially, not D50 as previously proposed. Unexpectedly, the salt bridge interaction between E117 and the −4 position of the retrieval signal makes only a limited contribution to binding affinity and does not explain the higher affinity for HDEL. The higher affinity for HDEL is due to the stronger π-π interaction between the histidine of the −4 position of the retrieval signal and W120 in the receptor.

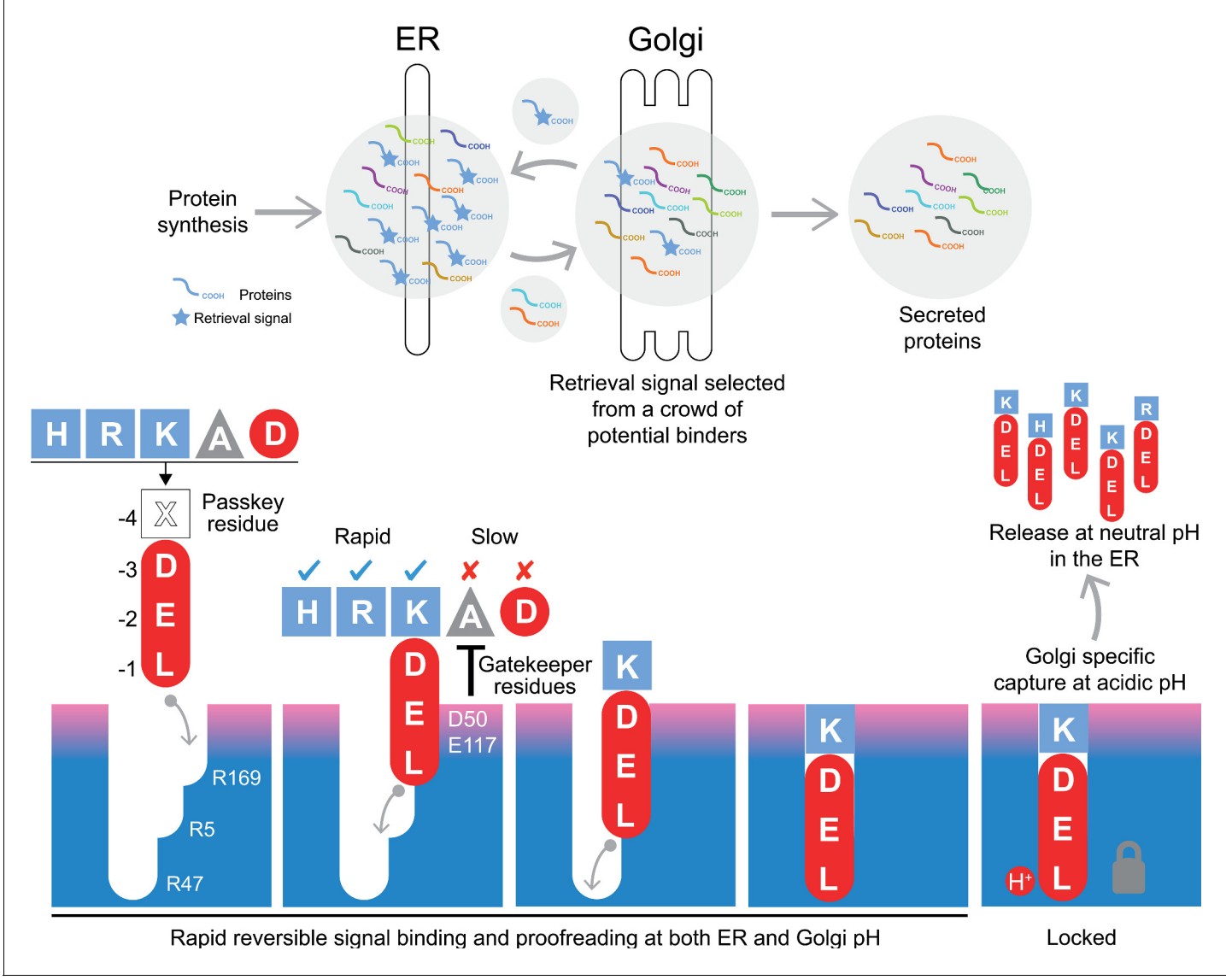

**Figure 8.** A combined proofreading and relay handover model for signal capture by the KDEL receptor. Newly synthesised secretory and ER luminal proteins are translocated into the ER and on to the Golgi. Those proteins with C-terminal retrieval signals are captured by the KDELR receptor and returned to the ER. Other proteins with different C-terminal sequences move on to be secreted. The retrieval signal can be broken down into two sections: the variable −4 passkey position and the −1 to −3 positions with free carboxyl-terminus. Signals are initially captured through their free carboxyl-terminus by the receptor R169. This is then handed over to R5 and finally R47 in a relay mechanism. Sequences are proofread for the residue at the −4 position by gatekeeper residues D50 and E117. Unwanted signal variants are rejected. Only signals that completely enter the binding pocket and engage R47 can undergo pH dependent capture and return to the ER.

However, because E117A mutant receptors have expanded specificity and can recognise ADEL, we conclude that the side chain at the −4 position is unlikely to play a major role in binding affinity for signals other than HDEL. For these reasons, we refer to the −4 position as the passkey residue, important for selection and entry of the signal. By determining net charge on the signal, it may thus play a greater role in initial binding kinetics.

Taken together, these data support a model for retrieval sequence recognition that explains both the importance of the free C-terminal carboxyl group and how changes at the −4 position can modulate binding to the receptor.

## Upstream residues and retrieval specificity of KDEL receptor variants

There are some unanswered questions, predominantly related to the role of the −5 and −6 positions of retrieval signals, and the properties and function of KDELR3. Previous work has suggested that the −5 and −6 positions of the retrieval signal also play a key role in signal binding (*Alanen et al., 2011*), and that the individual human KDEL receptors have slightly different specificities (*Raykhel et al., 2007*). However, these properties are not completely consistent with the structures, pattern of sequence conservation, or wider analysis presented here. It is noteworthy that both those studies used a bimolecular fluorescence complementation approach where the signal and receptor are dimerised by a split YFP molecule that will likely contribute to the observed signal binding affinity. This will interfere with the initial proofreading mechanism described here, making comparison with our data difficult. Based on the structures it seems reasonable that the −5 position may contribute to signal proofreading in some cases. However, as we show, a wide variety of signals that lack any obvious conserved features upstream of the canonical tetrapeptide function efficiently to trigger ER retrieval of the receptor (*Figure 1a* and S5b-S5e), suggesting the −5 position modulates but does not play an essential role in signal recognition. Although the structures show that the mode of signal recognition involves residues conserved in all three human KDEL receptors, in KDELR3 the loop between helices 4 and 5 upstream of E117 is altered in sequence compared to KDELR1 and R2 (*Figure 5a*). This is close to other residues on the surface of the receptor important for signal selectivity. Previous work has indicated that KDELR3 is upregulated under stress conditions and may be more selective for HDEL signals than KDELR1 and KDELR2 (*Raykhel et al., 2007*; *Trychta et al., 2018*). The precise consequences of these differences and specific functional roles for KDELR3 remain unclear, and it thus deserves further investigation including new structural data.

## Signal variants increase the dynamic range of the ER retrieval system

One important outcome from our work is the idea that KDEL receptors are not optimised for an individual signal and must retain the ability to differentiate variant high and low affinity ER retrieval signals. We propose that cells exploit these properties to maximise the retrieval efficiency of a broad range of ER resident proteins with widely different abundance, over 2 or 3 orders of magnitude. This idea provides an explanation for the functional significance of the affinity differences of retrieval signal variants in mammalian cells. The most abundant proteins use the KDEL retrieval signal, whereas lower abundance proteins tend to carry the HDEL signal. By artificially increasing the concentration of HDEL proteins, we can show that this effectively poisons the ER retrieval system, leading to the secretion of normally retained ER chaperones. This behaviour is reminiscent of other cellular regulatory systems, where substrate or signal binding properties are optimised for rate and turnover, rather than for the highest affinity which can reduce throughput of the pathway. Indeed, in some cases electrostatic properties are exploited to create rapid-binding high-affinity inhibitors that outcompete substrates (*Cundell et al., 2016*; *Schreiber and Fersht, 1996*). This may explain why histidine has been selected for the highest affinity variant of the signal to counteract this effect. For the HDEL variant, protonation of both histidine 12 in the receptor and histidine at the −4 position of the retrieval signal favour binding to the receptor in the Golgi. However, deprotonation of both the retrieval signal and receptor at pH 7.0 enable rapid release in the ER, and hence receptor recycling to the Golgi. Thus, HDEL binds more tightly than KDEL in the Golgi, but still releases rapidly in the ER. A signal with the same affinity as HDEL that was not proton dependent would strongly inhibit retrieval even at low concentration due to slow release at neutral pH. An alternative mechanism to capture low abundance ER proteins would have been to increase the cellular concentration of the KDEL receptor from the observed low levels. That would require receptors to be nearly stoichiometric with cargo, a problematic proposition considering the millimolar concentration of ER chaperones. These potential traps are avoided by the combination of pH-regulation of both the receptor and the high-affinity HDEL signal. Thus, the versatile binding site architecture of the KDEL receptor enables differentiation of both high and low affinity signals, thereby enabling efficient ER retrieval of both low and high abundance proteins in eukaryotic cells.

# Materials and methods

**Key resources table**

| Reagent type (species) or resource | Designation | Source or reference | Identifiers | Additional information |
|---|---|---|---|---|
| Strain, strain background (*Escherichia coli*) | XL1-Blue Competent Cells | Agilent Technologies | 200249 | Used to prepare plasmid DNA |
| Strain, strain background (*Saccharomyces cerevisiae*) | Bj5460 | ATCC | 208285 | Used for KDELR protein expression |
| Cell line (*African green monkey*) | COS-7 kidney fibro blast-like cell line | ATCC | CRL-1651 | ER retrieval assays |
| Cell line (*Homo-sapiens*) | HeLa S3 cervical adenocarcinoma | ATCC | CCL-2.2 | Protein secretion assays |
| Antibody | TGN46 sheep polyclonal | Bio-rad (AbD Serotec) | AHP500G | IF (1:1000) |
| Antibody | GRP78 BiP rabbit polyclonal | Abcam | ab21685 | WB (1:1000) |
| Antibody | PDI rabbit polyclonal | ProteinTech | #11245–1 | WB (1:1000) |
| Antibody | ERp72 rabbit monoclonal | Cell Signalling Technology | #5033S | WB (1:1000) |
| Antibody | ERp44 rabbit monoclonal | Cell Signalling Technology | # 3798S | WB (1:1000) |
| Antibody | KDEL receptor mouse monoclonal | Enzo Life Sciences | ADI-VAA-PT048 | IF (1:1000) WB (1:1000) |
| Antibody | RFP mouse monoclonal | Chromotek | 6G6 | WB (1:1000) Detects mScarlet on Western blot |
| Antibody | Donkey anti-Mouse IgG Alexa Fluor 488 | Invitrogen | A-21202 | IF (1:2000) Secondary |
| Antibody | Donkey anti-Sheep IgG Alexa Fluor 647 | Invitrogen | A-21448 | IF (1:2000) Secondary |
| Antibody | Peroxidase-AffiniPure Donkey Anti-Rabbit IgG | Jackson Immuno Research | 711-035-152-JIR | WB (1:2000) Secondary |
| Antibody | Peroxidase-AffiniPure Donkey Anti-Mouse IgG | Jackson Immuno Research | 711-035-152-JIR | WB (1:2000) Secondary |
| Antibody | Peroxidase-AffiniPure Donkey Anti-Sheep IgG | Jackson Immuno Research | 713-035-147-JIR | WB (1:2000) Secondary |
| Recombinant DNA reagent | pcDNA3.1 hGHss-mScarlet -*H. sapiens* BiP$_{639-654}$ | *Bräuer et al., 2019* | KDEL$^{SEC}$ | PMID:30846601 |
| Recombinant DNA reagent | pcDNA3.1 hGHss-mScarlet -*H. sapiens* BiP$_{639-654}$ K651R | This paper | RDEL$^{SEC}$ | Material and methods. Available from Barr lab |
| Recombinant DNA reagent | pcDNA3.1 hGHss-mScarlet -*H. sapiens* BiP$_{639-654}$ K651H | This paper | HDEL$^{SEC}$ | Material and methods. Available from Barr lab |
| Recombinant DNA reagent | pcDNA3.1 hGHss-mScarlet -*H. sapiens* BiP$_{639-654}$ K651A (ADEL$^{SEC}$) | This paper | ADEL$^{SEC}$ | Material and methods. Available from Barr lab |
| Recombinant DNA reagent | pcDNA3.1 hGHss-mScarlet -*H. sapiens* BiP$_{639-654}$ K651D (DDEL$^{SEC}$) | This paper | DDEL$^{SEC}$ | Material and methods. Available from Barr lab |
| Recombinant DNA reagent | pcDNA3.1 hGHss-mScarlet -*S. cerevisiae* BiP$_{667-682}$ | This paper | Yeast BiP | Material and methods. Available from Barr lab |
| Recombinant DNA reagent | pcDNA3.1 hGHss-mScarlet -*S. pombe* BiP$_{648-663}$ | This paper | *S. pombe* BiP | Material and methods. Available from Barr lab |
| Recombinant DNA reagent | pcDNA3.1 hGHss-mScarlet -*S. pombe* BiP$_{648-663}$ A660K | This paper | *S. pombe* BiP A > K | Material and methods. Available from Barr lab |

*Continued on next page*

*Continued*

| Reagent type (species) or resource | Designation | Source or reference | Identifiers | Additional information |
|---|---|---|---|---|
| Recombinant DNA reagent | pcDNA3.1 hGHss-mScarlet -*K. lactis* BiP$_{664-679}$ | This paper | K. lactis BIP | Material and methods. Available from Barr lab |
| Recombinant DNA reagent | pcDNA3.1 hGHss-mScarlet -*K. lactis* BiP$_{664-679}$ D676K | This paper | K.lactis BiP D > K | Material and methods. Available from Barr lab |
| Recombinant DNA reagent | pcDNA3.1 hGHss-mScarlet -*H. sapiens* FKBP7$_{207-222}$ | This paper | FKBP7 | Material and methods. Available from Barr lab |
| Recombinant DNA reagent | pcDNA3.1 hGHss-mScarlet -*H. sapiens* FKBP9$_{555-570}$ | This paper | FKBP9 | Material and methods. Available from Barr lab |
| Recombinant DNA reagent | pcDNA3.1 hGHss-mScarlet -*H. sapiens* FKBP10$_{567-582}$ | This paper | FKBP10 | Material and methods. Available from Barr lab |
| Recombinant DNA reagent | pcDNA3.1 hGHss-mScarlet -*H. sapiens* FKBP14$_{196-211}$ | This paper | FKBP14 | Material and methods. Available from Barr lab |
| Recombinant DNA reagent | pEF5/FRT human KDELR1-GFP | *Bräuer et al., 2019* | KDELR1 | PMID:30846601 |
| Recombinant DNA reagent | pEF5/FRT human KDELR2-GFP | This paper | KDELR2 | Material and methods. Available from Barr lab |
| Recombinant DNA reagent | pEF5/FRT human KDELR1 H12A-GFP | *Bräuer et al., 2019* | H12A | Expression in mammalian cells for functional assays; PMID:30846601 |
| Recombinant DNA reagent | pEF5/FRT human KDELR1 D50N-GFP | This paper | D50N | Material and methods. Available from Barr lab |
| Recombinant DNA reagent | pEF5/FRT human KDELR1 N54K-GFP | This paper | N54K | Material and methods. Available from Barr lab |
| Recombinant DNA reagent | pEF5/FRT human KDELR1 N54R-GFP | This paper | N54R | Material and methods. Available from Barr lab |
| Recombinant DNA reagent | pEF5/FRT human KDELR1 E117Q-GFP | This paper | E117Q | Material and methods. Available from Barr lab |
| Recombinant DNA reagent | pEF5/FRT human KDELR1 E117D-GFP | This paper | E117D | Material and methods. Available from Barr lab |
| Recombinant DNA reagent | pEF5/FRT human KDELR1 E117A-GFP | This paper | E117A | Material and methods. Available from Barr lab |
| Recombinant DNA reagent | pEF5/FRT human KDELR1 E117N-GFP | This paper | E117N | Material and methods. Available from Barr lab |
| Recombinant DNA reagent | pEF5/FRT human KDELR1 W120F-GFP | This paper | W120F | Material and methods. Available from Barr lab |
| Recombinant DNA reagent | pEF5/FRT human KDELR1 W120A-GFP | This paper | W120A | Material and methods. Available from Barr lab |
| Recombinant DNA reagent | pEF5/FRT human KDELR1 R169K-GFP | This paper | R169K | Material and methods. Available from Barr lab |
| Recombinant DNA reagent | pEF5/FRT human KDELR1 R169A-GFP | This paper | R169A | Material and methods. Available from Barr lab |
| Recombinant DNA reagent | pEF5/FRT human KDELR1 E117A/W120A-GFP | This paper | E117A/W120A | Material and methods. Available from Barr lab |
| Recombinant DNA reagent | pEF5/FRT human KDELR1 D50N/N54K-GFP | This paper | D50N/N54K | Material and methods. Available from Barr lab |
| Recombinant DNA reagent | pEF5/FRT human KDELR1 D50N/N54R-GFP | This paper | D50N/N54R | Material and methods. Available from Barr lab |
| Recombinant DNA reagent | pEF5/FRT human KDELR1 D50N/E117Q-GFP | This paper | D50N/E117Q | Material and methods. Available from Barr lab |
| Recombinant DNA reagent | pEF5/FRT human KDELR1 D50N/E117N-GFP | This paper | D50N/E117N | Material and methods. Available from Barr lab |

*Continued on next page*

*Continued*

| Reagent type (species) or resource | Designation | Source or reference | Identifiers | Additional information |
|---|---|---|---|---|
| Recombinant DNA reagent | pEF5/FRT human KDELR1 N54K/E117Q-GFP | This paper | N54K/E117Q | Material and methods. Available from Barr lab |
| Recombinant DNA reagent | pEF5/FRT human KDELR1 D50N/N54K/E117Q-GFP | This paper | D50N/N54K/E117Q | Material and methods. Available from Barr lab |
| Recombinant DNA reagent | pEF5/FRT human KDELR1 N54R/E117N-GFP | This paper | N54R/E117N | Material and methods. Available from Barr lab |
| Recombinant DNA reagent | pEF5/FRT human KDELR1 D50N/N54R/E117N-GFP | This paper | D50N/N54R/E117N | Material and methods. Available from Barr lab |
| Recombinant DNA reagent | pEF5/FRT human KDELR1 D50N/N54K/E117Q/W120A-GFP | This paper | D50N/N54K/E117Q/W120A | Material and methods. Available from Barr lab |
| Recombinant DNA reagent | pEF5/FRT human KDELR1 D50N/N54R/E117N/W120A-GFP | This paper | D50N/N54R/E117N/W120A | Material and methods. Available from Barr lab |
| Recombinant DNA reagent | pDDGFP-Leu2d-GgKDELR2 | Addgene | 123618 | Protein expression in yeast for biochemical assays and structures |
| Recombinant DNA reagent | pDDGFP-Leu2d-GgKDELR2_H12A | This paper | KDELR2_H12A | Material and methods. Available from Newstead lab |
| Recombinant DNA reagent | pDDGFP-Leu2d-GgKDELR2_E117A | This paper | KDELR2_E117A | Material and methods. Available from Newstead lab |
| Recombinant DNA reagent | pDDGFP-Leu2d-GgKDELR2_E117D | This paper | KDELR2_E117D | Material and methods. Available from Newstead lab |
| Recombinant DNA reagent | pDDGFP-Leu2d-GgKDELR2_E117Q | This paper | KDELR2_E117Q | Material and methods. Available from Newstead lab |
| Recombinant DNA reagent | pDDGFP-Leu2d-GgKDELR2_E127A | This paper | KDELR2_E127A | Material and methods. Available from Newstead lab |
| Recombinant DNA reagent | pDDGFP-Leu2d-GgKDELR2_E127Q | This paper | KDELR2_E127Q | Material and methods. Available from Newstead lab |
| Recombinant DNA reagent | pDDGFP-Leu2d-GgKDELR2_W120A | This paper | KDELR2_W120A | Material and methods. Available from Newstead lab |
| Recombinant DNA reagent | pDDGFP-Leu2d-GgKDELR2_W120F | This paper | KDELR2_W120F | Material and methods. Available from Newstead lab |
| Recombinant DNA reagent | pDDGFP-Leu2d-GgKDELR2_R169A | This paper | KDELR2_R169A | Material and methods. Available from Newstead lab |
| Recombinant DNA reagent | pDDGFP-Leu2d-GgKDELR2_R169K | This paper | KDELR2_R169K | Material and methods. Available from Newstead lab |
| Peptide, recombinant protein | TEV Protease | Merck | T4455-10KU | |
| Peptide, recombinant protein | $^3$H-TAEHDEL | Cambridge Research Biochemicals | custom synthesis | 185 MBq 106 Ci/mmol |
| Peptide, recombinant protein | $^3$H-TAEKDEL | Cambridge Research Biochemicals | custom synthesis | 185 MBq 128 Ci/mmol |
| Peptide, recombinant protein | TAEHDEL | Cambridge peptides | custom synthesis | |
| Peptide, recombinant protein | TAEKDEL | Cambridge peptides | custom synthesis | |

Continued

| Reagent type (species) or resource | Designation | Source or reference | Identifiers | Additional information |
|---|---|---|---|---|
| Peptide, recombinant protein | TAERDEL | Cambridge peptides | custom synthesis | |
| Peptide, recombinant protein | TAEDDEL | Cambridge peptides | custom synthesis | |
| Peptide, recombinant protein | TAEKDEL-CONH | Cambridge peptides | custom synthesis | C-amidated peptide variant. |
| Peptide, recombinant protein | TAEHDEL-CONH | Cambridge peptides | custom synthesis | C-amidated peptide variant. |
| Chemical compound, drug | Sodium phosphate monobasic ($NaH_2PO_4$) | Sigma | S8282 | |
| Chemical compound, drug | Sodium phosphate dibasic ($Na_2HPO_4$) | Sigma | 71640 | |
| Chemical compound, drug | Sodium periodate ($NaIO_4$) | Sigma | 311448 | |
| Chemical compound, drug | 16% (w/v) Formaldehyde | Thermo Fisher Scientific | 28908 | |
| Chemical compound, drug | Saponin | Sigma | S7900 | |
| Chemical compound, drug | L-Lysine monohydrochloride | Sigma | 62929 | |
| Chemical compound, drug | Mowiol 4–88 | Millipore | 475904 | |
| Chemical compound, drug | Trichloroacetic acid | Sigma | T6399 | |
| Chemical compound, drug | Dodecyl maltoside (DDM) | Glycon | D97002-C | |
| Chemical compound, drug | Cholesteryl hemisuccinate (CHS) | Sigma | C6512 | |
| Chemical compound, drug | Monoolein | Sigma | M7765 | |
| Software, algorithm | Metamorph 7.5 | Molecular Dynamics Inc | http://www.moleculardevices.com | Microscope image data acquisition |
| Software, algorithm | Fiji 2.0.0-rc-49/1.52i | NIH Image | http://fiji.sc/ | Microscope image data analysis |
| Software, algorithm | GraphPad Prism 7 | GraphPad Software | http://www.graphpad.com | Graph plotting |
| Software, algorithm | R | R project for statistical computing | https://www.r-project.org | Statistical analysis and graph plotting |
| Software, algorithm | Adobe Illustrator CC | Adobe Systems Inc | http://www.adobe.com | Figure preparation |
| Software, algorithm | Adobe Photoshop CC | Adobe Systems Inc | http://www.adobe.com | Figure preparation |
| Software, algorithm | COOT | *Emsley et al., 2010* | https://www2.mrc-lmb.cam.ac.uk/personal/pemsley/coot | Macromolecular structure model building; PMID:20383002 |
| Software, algorithm | PyMOL | Schrodinger | https://pymol.org/2 | Molecular visualisation |
| Software, algorithm | Buster | Global Phasing | https://www.globalphasing.com | Structure refinement |

*Continued on next page*

*Continued*

| Reagent type (species) or resource | Designation | Source or reference | Identifiers | Additional information |
|---|---|---|---|---|
| Software, algorithm | GROMACS | *Abraham et al., 2015* | https://www.gromacs.org | Molecular dynamics |
| Software, algorithm | GMX_lipid17.ff: GROMACS | *Wu and Biggin, 2020* | http://doi.org/10.5281/zenodo.3610470 | Port of the Amber LIPID17 force field |
| Software, algorithm | MDAnalysis 1.0 | SciPy2016 | https://conference.scipy.org/proceedings/scipy2016/oliver_beckstein.html | Analysis of molecular dynamics simulations |
| Software, algorithm | Modeller 9.21 | *Webb and Sali, 2016* | https://salilab.org/modeller/ | PMID:27322406 |
| Other | Ultima Gold Scintillation Fluid | Perkin Elmer | 6013326 | |
| Other | HisPurTM | Thermo Fisher Scientific | 25214 | |
| Other | HisTrap HP | Cytiva | 17-5248-01 | |
| Other | Superdex 200 10/300 GL | Cytiva | 28-9909-44 | |
| Other | Ultra-15 Centrifugal Filter Unit, 50K NMWC | Amicon | UFC905024 | |
| Other | Yeast Drop Out media -Ura | Formedium | DCS0169 | |
| Other | Yeast Drop Out media -Leu | Merck | Y1376-20G | |
| Other | Tunair Flasks | Sigma | Z710822-4EA | |
| Other | Dulbecco's modified Eagle's medium | Thermo Fisher Scientific | 31966–047 | |
| Other | Foetal Bovine Serum | Sigma | F9665 | |
| Other | TrypLE Express Enzyme | Thermo Fisher Scientific | 12605036 | |
| Other | Opti-MEM | Thermo Fisher Scientific | 11058021 | |
| Other | EZ-PCR Mycoplasma Test Kit | Geneflow | K1-0210 | |
| Other | TransIT-LT1 | Mirus Bio LLC | MIR 2306 | |
| Other | ECL western blotting reagent | Cytiva | RPN2106 | |

## Mammalian cell lines

African green monkey fibroblast-like kidney COS-7 cells (ATCC #CRL-1651) and human cervical adenocarcinoma HeLa cells (ATCC #CCL-2.2 confirmed by STR profiling) were cultured in DMEM (Invitrogen, Thermo Fisher Scientific) containing 10% [vol/vol] foetal bovine serum (Sigma) at 37°C and 5% $CO_2$. For passaging, cells were washed once in PBS, and then removed from the dish by 5 min incubation with TripLE Express (Thermo Fisher Scientific). Mycoplasma negative status of cell lines was confirmed using the EZ-PCR Mycoplasma Test Kit with internal control (K1-0210, Geneflow).

## ER retrieval assays

To create KDELR1-GFP and KDELR2-GFP, the reading frames for *Homo sapiens* KDELR1 (Uniprot: P24390) and KDELR2 (Uniprot: P33947) were cloned into the pEF5/FRT low level mammalian expression vector with a C-terminal 20 amino acid linker made up of 5 copies of Gly-Ser-Ser-Ser followed by GFP. Specific point mutations were introduced using the Quickchange protocol (Stratagene) and are described in the key resources table and figures. To create the mScarlet-KDEL[sec] ligand construct, mScarlet with an N-terminal hGH signal peptide and the 16 C-terminal residues of human BiP at its C-terminus, containing the KDEL signal, was cloned into the pcDNA3.1 vector. This was then modified using site-directed mutagenesis or annealed oligo ligation to create C-terminal

retrieval signal variants from known human and yeast ER proteins. For ER retrieval assays, COS-7 cells were grown on 10 mm diameter 0.16–0.19 mm thick glass coverslips in DMEM containing 10% [vol/vol] bovine calf serum at 37°C and 5% $CO_2$. Cells were plated at 50,000 cells per well of a 6-well plate, each well containing two coverslips. The cells were transfected after 24 hr with 0.25 μg KDELR1-GFP or KDELR2-GFP and 0.5 μg mScarlet-ligand (+xDEL ligand) or 0.25 μg KDELR-GFP and 0.5 μg pcDNA3.1 (− ligand) diluted in 100 μl Opti-MEM and 3 μl TransIT LT1 (Mirus Bio LLC). After a further 18 hr, cells were washed twice with 2 mL of PBS, then fixed for 2 hr in 2 mL 2% wt/vol) formaldehyde in 87.5 mM lysine, 10 mM sodium periodate, buffered with 87.5 mM sodium phosphate pH 7.4. Subsequently, coverslips were washed three times in 2 mL permeabilisation solution 100 mM sodium phosphate pH 7.4, then permeabilised in 1 mg mL$^{-1}$ BSA, 0.12 mg mL$^{-1}$ saponin, and 100 mM sodium phosphate pH 7.4 for 30 min. Primary and secondary antibody staining was performed sequentially for 60 min in permeabilisation solution at 22°C, with three washes with 2 mL PBS in between. The Golgi marker protein TGN46 was detected by antibody (sheep anti-TGN46 AHP500G; AbD Serotec) and an Alexa 647 conjugated secondary anti-sheep secondary (A-21448, Invitrogen). In *Figure 1d and e*, endogenous KDELR was detected by antibody (mouse anti-KDELR ADI-VAA-PT048; Enzo Life Sciences) and Alexa 488 conjugated anti-mouse secondary antibody (A-21202, Invitrogen). For retrieval assays in other figures, KDELR-GFP fusion proteins were directly detected by fluorescence. The mScarlet-xDEL fusion proteins were directly detected by fluorescence. Coverslips were mounted on glass slides in Mowiol 4–88 and imaged with a 60×/1.35 NA oil immersion objective on an Olympus BX61 upright microscope (with filtersets for DAPI, GFP/Alexa-488,–555, −568, and −647 (Chroma Technology Corp.), a 2048 × 2048 pixel CMOS camera (PrimΣ; Photometrics), and MetaMorph 7.5 imaging software (Molecular Dynamics Inc). Illumination was provided by a wLS LED illumination unit (QImaging). Image stacks of 3–5 planes with 0.3 μm spacing through the ER and Golgi were taken. The image stacks were then maximum intensity projected and the selected channels merged to create 24-bit RGB TIFF files in MetaMorph. To produce the figures, images in 24-bit RGB format were cropped in Photoshop to show individual cells and then placed into Illustrator (Adobe Systems Inc). The signal for the KDEL receptor (integrated pixel intensity) was measured in individual cells using FIJI (*Schindelin et al., 2012*) for the Golgi region defined by the TGN46 Golgi marker and for the entire cell in the presence (+) and absence (-) of ligand. The fraction of KDEL receptor in the Golgi apparatus was calculated by dividing the Golgi signal by the total cell signal. These combined single cell data were then used for the statistical analysis of ER retrieval in R.

To estimate the effect sizes and significance of receptor mutations for ligand-mediated ER retrieval, pooled data was analysed in R using the open-source package dabestr (*Ho et al., 2019*; *R Core Development Team, 2017*; *Wickham, 2010*). Data are presented as Cumming estimation plots, where the raw data is plotted on the upper axes and mean differences are plotted as bootstrap sampling distributions on the lower axes for 5000 bootstrap samples. Each mean difference is depicted as a dot. Each 95% confidence interval is indicated by the ends of the vertical error bars; the confidence interval is bias-corrected and accelerated. The p values reported are the likelihood of observing the effect size, if the null hypothesis of zero difference is true. For each permutation p value, 5000 reshuffles of the control and test labels were performed.

## ER secretion assays

For ER chaperone secretion assays, HeLa S3 cells were transfected with 0.5 μg mScarlet-ligand (+xDEL ligand) or 0.5 μg pcDNA3.1 (− ligand), and allowed to express the respective proteins for 24 hr. The media were TCA precipitated and both cell and media were resuspended and boiled in SDS-PAGE sample buffer. All samples were analysed by Western blotting (Trans-Blot Turbo transfer system, Bio-Rad) for xDEL ligand (mouse anti-RFP 6G6, Chromotek), resident ER chaperones BIP (rabbit #ab21685, Abcam), PDI (rabbit #11245–1, ProteinTech), ERP72 (rabbit #5033S, Cell Signalling Technology), ERP44 (rabbit #3798S, Cell Signalling Technology) and the KDEL receptor (mouse ADI-VAA-PT048, Enzo Life Sciences). HRP-conjugated secondary antibodies and the ECL reagent were used to detect signals on film. Films were then digitised and signals measured by densitometry in FIJI (*Schindelin et al., 2012*). Data were plotted as bar graphs in GraphPad Prism.

## KDEL receptor crystallisation and structure determination

*Gg* KDELR2 was expressed and purified as described previously (*Bräuer et al., 2019*), concentrated to 14.5 mg mL$^{-1}$ and incubated with 6.4 mM TAEHDEL or RDEL peptide on ice for one hour prior to crystallisation. Crystals were set up at 20°C as above using precipitant 30% (v/v) PEG 600, 100 mM MES pH 6.0, 100 mM Sodium Nitrate. Phases were determined via molecular replacement using Phaser and employing PDB:6I6H as the search model with the TAEKDEL peptide removed from the search model. The TAEHDEL and TAERDEL peptides were built into difference density using Coot (*Emsley et al., 2010*), followed by refinement in BUSTER (*Blanc et al., 2004*).

## Retrieval signal binding assays

Binding assays were performed in 20 mM MES pH 5.4, 40 mM Sodium Chloride, 0.01% DDM 0.0005% CHS unless stated otherwise. Five µL of $^{3}$H-TAEKDEL or $^{3}$H-TAEHDEL (Cambridge peptides, UK) at 20 nM were incubated with 5 µL of *Gg* KDELR or variants thereof at the desired concentration at 20°C for 10 min. The reaction was then filtered through a 0.22 µm mixed cellulose ester filter (Millipore, USA) using a vacuum manifold. Filters were then washed with 2 × 500 µL buffer. The amount of bound peptide was measured using scintillation counting in Ultima Gold (Perkin Elmer). Experiments were performed a minimum of three times to generate an overall mean and standard deviation. Data was normalised to the maximal binding at pH 5.4 and fit with a four-parameter logistic non-linear regression model.

## Thermal stability measurements

Concentrated (~10 mg ml) GgKDELR2 was diluted to 0.2 mg/ml into buffer consisting of 10 mM citric acid, 20 mM di-sodium phosphate at pH 5.4, 5.9, 6.4, or 7.0 containing 0.01% DDM:CHS (20:1 ratio). To this 0.5 mM KDEL, RDEL, or HDEL peptide (diluted in water) was added, or water as a control. The sample was incubated at room temperature for 15 min. Thermal measurements were carried out in a range from 20°C to 90°C with 1°C/min steps using a Prometheus NT.48. The PR. ThermControl (NanoTemper) software was used to calculate the melting temperature for each condition. The data shown in the manuscript is the calculated melting temperature for each peptide at the given pH with the T$_m$ for the water control at the same pH subtracted.

## Relative binding free energy calculations

To compute the free energy of the deprotonation of the histidine or lysine and the mutation of lysine to histidine, molecular-mechanics-based alchemical transformation was performed. The free energy difference was taken as the difference in the free energy of the transformation between the protein-peptide complex and the peptide in solution. The KDEL receptor in the protein-peptide complex was taken from the crystal structure (KDEL: 6I6B *Bräuer et al., 2019*; HDEL: 6Y7V). The C-terminus of the receptor was modelled to full length using Modeller 9.21 *Webb and Sali, 2016*; 100 models were created and the one with the best DOPE score was selected (*Shen and Sali, 2006*). The protein was then embedded into a lipid membrane containing 186 DMPC lipids using the procedure described by us previously (*Wu et al., 2019*). The system of peptide in solution was constructed by taking the coordinates of the peptide from the crystal structure and placing in a box, where the box edge was at least 2 nm from the peptide. Both systems were solvated and neutralised to final salt concentration of 150 mM NaCl. For the deprotonation calculations, the change in charge in the system was counteracted by simultaneously charging a sodium ion in the corner of the box (i.e. at the start of the process the charge was zero and by the end it was +1). To minimise the interactions between the histidine (or lysine) and this alchemical sodium ion, the histidine/lysine residues were restrained to the centre of the box via their Cα atom using a harmonic restraint of 1000 kJ/mol/nm$^2$ and the alchemical sodium ion was either restrained to the edge of the box for the peptide in solution or restrained to the z-axis in the case of the peptide-protein complex.

The Amber ff14SB force field (*Maier et al., 2015*) was used to describe the protein and alchemical transformation was done with pmx (*Gapsys et al., 2015*). Lipids were described by LIPID17, which was ported from Amber to GROMACS by us (GMX_lipid17.ff: GROMACS. Port of the Amber LIPID17 force field. Zenodo. http://doi.org/10.5281/zenodo.3610470). The simulations were run with GROMACS 2018 (*Abraham et al., 2015*). The simulation input parameters were set according to recommendations suggested by pmx. Since the equilibrium method was used, the sc-alpha and sc-

sigma parameters were set to 0.5 and 0.3 respectively. For the lysine to histidine transformation, a total of 21 lambda windows with 0.05 equal spacing were used to transform the charge and the vdw parameters at the same time. A soft-core potential was used for the coulombic interactions to avoid singularity effects. For the deprotonation calculations, 11 equally spaced windows were used to change the partial charge and an addition window was used to complete the transformation. After energy minimisation, each window was run for 200 ps in the NVT ensemble and one ns in an NPT ensemble with positional restraints of 1000 kJ/mol/nm$^2$ to reach a final temperature of 310 K and 1 bar. 30 ns production runs with replicate exchanges at intervals of 1 ps were then performed. Data were analysed using the Multistate Bennett Acceptance Ratio with alchemical analysis within the first five ns discarded (*Klimovich et al., 2015*). For each transformation, three replicates were performed and the result presented as the mean and standard deviation. For the LYS/HIP transformation, since both HDEL-bound and KDEL-bound structure were available, six simulations (three starting from KDEL-bound structure and three from HDEL-bound structure) were used to produce the results.

To compute the free energy difference of KDEL to HDEL transformation, the total free energy difference of alchemically changing KDEL to HDEL is computed as

$$\Delta G_{KDEL \rightarrow HDEL} = \Delta G_{LYS \rightarrow LYS/N} + \Delta G_{LYS \rightarrow HIP} - \Delta G_{HIP \rightarrow HIP/D/E}$$

where $\Delta G_{HIP \rightarrow LYS}$ is the free energy difference of converting KDEL to HDEL when both lysine and histidine are in the protonated form. $\Delta G_{HIP \rightarrow HIP/D/E}$ and $\Delta G_{LYS \rightarrow LYS/N}$ is the free energy of converting protonated histidine or lysine from the protonated to an ensemble of protonated and deprotonated forms (for example we might calculate the energy to go from 100% protonated to an ensemble of 40% protonated and 60% deprotonated):

$$\Delta G_{LYS \rightarrow LYS/N} = w_{LYS}0 + w_{LYN} \left( \Delta G_{LYS \rightarrow LYN} - \Delta G_{LYS_{offset}} \right) - T\Delta S$$

$$\Delta G_{HIP \rightarrow HIP/D/E} = w_{HIP}0 + w_{HID} \left( \Delta G_{HIP \rightarrow HID} - \Delta G_{HIP_{offset}} \right) + w_{HIE} \left( \Delta G_{HIP \rightarrow HIE} - \Delta G_{HIP_{offset}} \right) - T\Delta S$$

The $\Delta G_{HIP_{offset}}$ and $\Delta G_{LYS_{offset}}$ are terms to calibrate the computational protonation free energy to the experimental microscopic pKa (histidine: 6.0; lysine: 8.95) and were defined as:

$$\Delta G_{offset} = 2.303RT \times (7.0 - pka)$$

$w$ is the Boltzmann weight of each protonation state and is computed as:

$w = \dfrac{e^{\frac{-\Delta G}{RT}}}{\sum e^{\frac{-\Delta G}{RT}}}$ and $\Delta S$ is the configurational entropy and is defined as:

$$\Delta S = -R \sum w \ln w$$

## Quantum mechanical calculations for the effect of HDEL protonation

To explore the interactions between the signal and receptor, the histidine of the HDEL signal and tyrosine (W120) of the receptor were taken from the crystal structure and capped at both ends (with acetyl and amide groups the N and C-termini, respectively). The hydrogens were added to the complex and the three different protonation sates of the histidine were constructed with Maestro 2019.2. The capped three amino acid complex were geometry minimised with non-hydrogen atoms constrained at the RI-B3LYP-D3(BJ)/def2-TZVP theory level with geometry counterpoise (*Grimme et al., 2010*; *Grimme et al., 2011*; *Kruse and Grimme, 2012*; *Weigend, 2006*; *Weigend and Ahlrichs, 2005*) using ORCA 4.2.0 (*Neese, 2012*). The interactions between the three different protonation states of the histidine and W120 were computed at the SAPT2+/jun-cc-pVDZ (*Parker et al., 2014*) theory level from the geometry optimised structure using psi4 1.3.2 (*Parrish et al., 2017*).

## Simulation of signal engagement with the binding site

To obtain a converged view of how the KDEL peptide enters the KDEL receptor, umbrella sampling was used to enhance the sampling of the behaviour of the C-terminus in the binding pocket. The initial frames were generated by pulling the N-terminus of the KDEL peptide out of the binding pocket using a moving restraint with GROMACS 2019.4/plumed 2.6.0 (*PLUMED consortium, 2019*). The

collective variable (CV) was defined as the distance between the N-terminus of the KDEL peptide (N atom) and the centre of the binding pocket, which was defined as the centre of the C$\alpha$ atoms of residue 9, 44, 64, 124, and 162. Pulling was performed using a CV = 1.8 nm to 3.3 nm with a restraint strength of 1000 kJ/mol/nm$^2$ for 100 ns. To prevent the complete dissociation of the peptide from the receptor, a one-side distance restraint was applied on the distance between the C-terminus of the peptide (atom C) and the binding pocket at 1.7 nm with a strength of 1000 kJ/mol/nm$^2$. Sixteen windows were set up where the CV was varied from 1.8 nm to 3.3 nm with a step of 0.1 nm and were run for 500 ns. The results were analysed with MDAnalysis 1.0 (https://conference.scipy.org/proceedings/scipy2016/oliver_beckstein.html).

## Quantification and statistical analysis

Details of the number of experimental repeats, numbers of cells analysed and the relevant statistics are detailed in the figure legends and specific method details.

## Acknowledgements

We thank the staff at I24 Diamond Light Source, UK for access to the beamline. This work was supported by a Wellcome Trust award (219531/Z/19/Z) to SN, FAB and PCB. Computation time was provided by JADE (EP/P020275/1) and ARCHER via HECBioSim (http://www.hecbiosim.ac.uk), supported by an EPSRC grant (EP/R029407/1) to PCB. PB and ZW were Wellcome Trust PhD students (109133/Z/15/A and 203741/Z/16/A).

## Additional information

### Funding

| Funder | Grant reference number | Author |
|---|---|---|
| Wellcome Trust | 219531/Z/19/Z | Philip C Biggin<br>Francis A Barr<br>Simon Newstead |
| Wellcome Trust | 203741/Z/16/A | Zhiyi Wu |
| Wellcome Trust | 109133/Z/15/A | Philipp Bräuer |
| Engineering and Physical Sciences Research Council | EP/R029407/1 | Philip C Biggin |

The funders had no role in study design, data collection and interpretation, or the decision to submit the work for publication.

### Author contributions

Andreas Gerondopoulos, Francis A Barr, Conceptualization, Formal analysis, Supervision, Funding acquisition, Investigation, Visualization, Methodology, Writing - original draft, Project administration, Writing - review and editing; Philipp Bräuer, Zhiyi Wu, Joanne L Parker, Formal analysis, Investigation, Visualization, Methodology, Writing - review and editing; Tomoaki Sobajima, Formal analysis, Investigation, Writing - review and editing; Philip C Biggin, Simon Newstead, Conceptualization, Data curation, Formal analysis, Supervision, Funding acquisition, Investigation, Visualization, Methodology, Writing - original draft, Project administration, Writing - review and editing

### Author ORCIDs

Philipp Bräuer  https://orcid.org/0000-0003-4127-2638
Tomoaki Sobajima  http://orcid.org/0000-0002-0753-5361
Zhiyi Wu  http://orcid.org/0000-0002-7615-7851
Philip C Biggin  https://orcid.org/0000-0001-5100-8836
Francis A Barr  https://orcid.org/0000-0001-7518-253X
Simon Newstead  https://orcid.org/0000-0001-7432-2270

Decision letter and Author response
Decision letter https://doi.org/10.7554/eLife.68380.sa1
Author response https://doi.org/10.7554/eLife.68380.sa2

# Additional files

**Supplementary files**

• Supplementary file 1. Crystallographic data collection statistics. Values in parentheses are for the highest resolution shell.

• Supplementary file 2. Free energy differences for HDEL protonation states and KDEL. The free energy difference is expressed in kcal/mol and the sum is the ensemble free energy difference between KDEL and HDEL. HID signifies that the histidine is protonated at δ nitrogen and HIE means that the histidine is protonated at ε nitrogen, while HIP means that both positions are protonated. The occupancy of protonation state is computed at pH seven and expressed as a percentage (%).

• Supplementary file 3. Cation-π and π-π contributions between receptor W120 and the −4 histidine in the retrieval signal. The cation-π interaction is the energy resulting from induction calculated at the level of sSAPT0/jun-cc-pVDZ. The π-π interaction is the sum of exchange and correlation energy. The sum is the sum of cation-π and π-π energy.

• Transparent reporting form

## Data availability

Atomic coordinates for the models have been deposited in the Protein Data Bank (PDB) under accession codes 6Y7V and 6ZXR. Data generated or analysed during this study are included in the manuscript and supporting files. Source data files have been provided for Figures 1, 1Sup1, 1Sup2, 2, 3, 3Sup1, 4, 4Sup1, 6, 6Sup1, 6Sup2, 7, and 7Sup1.

The following datasets were generated:

| Author(s) | Year | Dataset title | Dataset URL | Database and Identifier |
|---|---|---|---|---|
| Braeuer P, Newstead S | 2021 | Crystal structure of the KDEL receptor bound to HDEL peptide at pH 6.0 | https://www.rcsb.org/structure/6Y7V | RCSB Protein Data Bank, 6Y7V |
| Newstead S, Parker JL | 2021 | Crystal structure of the KDEL receptor bound to RDEL peptide at pH 6.0 | https://www.rcsb.org/structure/6ZXR | RCSB Protein Data Bank, 6ZXR |

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
