## [Decision Letter]

**Acceptance summary:**

Binding of cargo to sorting receptors in membrane trafficking is essential to cellular organization. This work is significant because it generates a detailed model of the key residues accounting for specificity and affinity of binding by the KDEL receptor. Interestingly, it is not the affinity per se that accounts for the specificity of cargo binding but rather charge-based exclusion of potentially competing signals. The work is meticulously carried out and the findings will likely be of significant interest to the field.

**Decision letter after peer review:**

Thank you for submitting your article "A baton-relay mechanism for ER retrieval signal capture and proofreading by the KDEL receptor" for consideration by *eLife*. Your article has been reviewed by 3 peer reviewers, one of whom is a member of our Board of Reviewing Editors, and the evaluation has been overseen by Olga Boudker as the Senior Editor. The following individual involved in review of your submission has agreed to reveal their identity: Edmund RS Kunji (Reviewer #2).

Essential revisions:

Overall the study was received with strong enthusiasm. The only major issue is a request for pH-dependent binding data, in particular for the RDEL peptide. The explanation from the reviewer is as follows:

1. The authors refer to the "acidic" Golgi versus the neutral pH of the ER. However, I think this is a wee bit misleading and would be more correct to refer to the mildy acidic Golgi vs neutral pH of the ER and give the pH values of 7.4 for ER and pH 6.2 for the Golgi lumen. This sets up the scientific question to be more nuanced, as its only a pH difference of 0.5 to 1.0 pH units, which has a profound influence on binding. Indeed, in the pH dependence of the HDEL peptide their is a largest shift in peptide binding between pH 6.4 and pH 7. This raises the question of how important is the pKa of the "K/R/H"-residues in relation to the pH differences? However, the authors have not shown the pH dependent profiles for the KDEL and RDEL. This is important to provide further evidence that the tighter binding of HDEL at the pH relevant differences (pH 6.2 and 7.4) is not because of its more optimal pKA rather than the additional pi-pi stacking to the tryptophan as conclude here. Furthermore, I think the RDEL is probably the most interesting to compare with, because arginine has such a high pKA (close to 14 DOI: 10.1002/pro.2647 ) and and so this residue will ALWAYS remain protonated. As such, one would expect the RDEL binding would been non-pH dependent. Could this difference have some functional consequence in comparison to the pH dependent H/K-DEL versions? However, if the RDEL variant retains some pH dependence, then this data would shed some light into the mechanism of signal recognition as it would suggest that their salt-bridge pairing of other residues could be important, e.g the pKa of E117 or D50 could be important.

Minor issues:

2. Computational methods were used to calculate an interact strength between the Trp and the various K/R/H"-residues in the -4 position. However, computational analysis was not carried out to calculate the pKA of the K/R/H"-residues, i.e., such with PROPKA as I think constant pH MD simulations would be beyond the scope of this paper. Related to the above question, I think it would be important to show the relationship between the charged state of the -4 residues in the peptide with pH-dependent binding affinities.

3. Related to the above 1 and 2 questions, is there any obvious difference with the proteins that use the RDEL sequence? For example, could these proteins be located in the Cis-Golgi wherein the pH difference is going to be less as compared to the pH of the TGN?

4. It was unclear to me whether the KDEL receptor was sufficient to cope with the turnover of abundant proteins, e.g., like BIP. One could imagine that under ER-stress that other isoforms such as KDEL3, for example, is required to help cope with trafficking. Related to this question, is it known what the preferences are between the K/R/HDEL variants and the KDELR1-3 variants?

5. R169 is interesting to me as the residues is located at the end of TMs and will therefore snorkel into the membrane bilayer. Do you think that the lipid composition differences between the Golgi and ER (high in PE) could also contribute to fine-tuning signal binding?

6. In my opinion, the terms "baton-relay" and "proof-reading" terminologies used imply more active processes than the simple biophysical interactions between the H/D/RDEL motif and the KDEL receptor.

7. Could tryptophan fluorescence be used as a proxy for peptide binding? Such an assay would make it easier to understand how peptide binding and dynamics are coupled.

8. Previous work has highlighted a role of D50 in the binding of the signal peptide, but the structures do not validate this claim, as the authors rightfully comment on. Still, closer inspection of the structures shows that D50 is at the same height as E117, which validated here to be an important residue, binding via ionic interactions to the positively charged residue of the signal sequences. Is it possible that the structures represent the end stage of the binding, a lower energy state. D50 could be another temporary entry point for the signal peptide, as stated in the text. If so, the mechanism could be equivalent to the insertion of a key, where the signal sequence binds to D50, but then rotates to E117 to lock the peptide in place, i.e. in the most stable binding pose, which could trigger the change in state leading to retrieval. The relay, made of three negatively charged groups on the signal sequence and three positively charged groups on the receptor, would still be important for driving the key into the lock. MD simulations can only probe a limited number of possible scenarios.

9. The authors note that they have used thermostability assays to study the folding of the receptor. My question is whether the binding of various signal peptides leads to differences in the thermostability of the receptor. Given the large number of ionic interactions involved in binding of the signal, it is certainly a possibility. It would provide a direct means to study the changes in stability of the receptor, which is related to aspects of energetics. If so, it might also be a route to study the interactions of the ER residents with the receptor.

---

## [Author Response]

Essential revisions:Overall the study was received with strong enthusiasm. The only major issue is a request for pH-dependent binding data, in particular for the RDEL peptide. The explanation from the reviewer is as follows:1. The authors refer to the "acidic" Golgi versus the neutral pH of the ER. However, I think this is a wee bit misleading and would be more correct to refer to the mildy acidic Golgi vs neutral pH of the ER and give the pH values of 7.4 for ER and pH 6.2 for the Golgi lumen. This sets up the scientific question to be more nuanced, as its only a pH difference of 0.5 to 1.0 pH units, which has a profound influence on binding. Indeed, in the pH dependence of the HDEL peptide their is a largest shift in peptide binding between pH 6.4 and pH 7. This raises the question of how important is the pKa of the "K/R/H"-residues in relation to the pH differences? However, the authors have not shown the pH dependent profiles for the KDEL and RDEL. This is important to provide further evidence that the tighter binding of HDEL at the pH relevant differences (pH 6.2 and 7.4) is not because of its more optimal pKA rather than the additional pi-pi stacking to the tryptophan as conclude here. Furthermore, I think the RDEL is probably the most interesting to compare with, because arginine has such a high pKA (close to 14 DOI: 10.1002/pro.2647 ) and and so this residue will ALWAYS remain protonated. As such, one would expect the RDEL binding would been non-pH dependent. Could this difference have some functional consequence in comparison to the pH dependent H/K-DEL versions? However, if the RDEL variant retains some pH dependence, then this data would shed some light into the mechanism of signal recognition as it would suggest that their salt-bridge pairing of other residues could be important, e.g the pKa of E117 or D50 could be important.

We have modified the text to refer to the Golgi as “mildly acidic”, which is certainly a more accurate description. One reason the KDEL system is so fascinating is the way it leverages a relatively small proton gradient to drive transport from the Golgi to the ER. Since pH is a log scale, this is a 10-fold difference in proton concentration in a range closely matched to the pKa of histidine. It isn’t only the difference in pH that is crucial, ER pH must not drop below 7, and Golgi pH should not rise above 6.2-6.4 or the system would fail to operate. Precise details of organelle pH regulation remain remarkably poorly understood, but as this comment shows are crucial to understand.

The pH-dependence of signal binding depends on both the receptor and the signal. For all three signal variants, the proton dependence of binding is mainly due to the receptor closing about the signal at Golgi pH. This is described in detail in our previous studies where we show that H12 in the receptor acts as a pH sensor and tunes a short hydrogen bond between Y158 and E127, and hydrogen bonds to R47, Y48, R159 and two water molecules to trap the C-terminus of the retrieval signal (Wu et al. 2020; PMC7547670) (Braüer et al. 2019; 30846601). Our new structures show the same arrangement for HDEL and RDEL, as our earlier work did for KDEL. In agreement with the idea binding is via the same mechanism, Figure 7f, 7g and Figure 7-supplement 1 show that the H12A mutant receptor does not bind KDEL or HDEL, or respond to KDEL, RDEL or HDEL.

For the signal there are differences for HDEL, when compared to KDEL and RDEL. We have added a new Figure 2-Supplement 1 including the pH dependence of RDEL, as well as HDEL and KDEL binding. Our previous work provides more analysis of the pH dependence of KDEL binding (Braüer et al. 2019). While the pH dependence of binding is similar for all three signals, HDEL always shows a higher affinity, no matter what assay is used, whereas KDEL and RDEL are similar. As we describe in the manuscript text, this is explained by interactions between the signal and W120 and depends on the protonation state of the histidine at position -4. Expected side chain pKas for lysine and arginine are above 10, whereas histidine is close to 6. Thus, HDEL will be protonated at Golgi pH and deprotonated at ER pH, whereas KDEL and RDEL will be protonated in both compartments. To investigate to what extent pH influences protonation state of the -4 position of the signal (rather than H12), we performed FEP calculations (shown in Author response table 1) on HDEL and KDEL. As the reviewer says, RDEL will certainly always be protonated and thus the more relevant calculation to compare the HDEL with is the KDEL signal. These calculations indicate two key things: 1) that the pKa of the -4 Histidine is shifted upward, thus confirming that the HDEL peptide will always be protonated when bound and 2) that the pKa is also influenced by the interaction with W120, completely consistent with our experimental data.

**Author response table 1. resptable1:** Calculated pKa values for retrieval signal -4 side chains.

	WT	W120F	W120A
HDEL	8.9+/-0.5	7.6+/-0.3	6.5+/-0.1
KDEL	11.1+/-0.4	9.7+/-0.2	9.0+/-0.1

Thus, the protonation/deprotonation equilibrium at the -4 position is not expected to modulate binding at ER or Golgi pH for R and KDEL. For HDEL, the pKa of 8.9 +/- 0.5 suggests there might be a small, yet important influence that would aid release of HDEL in the ER (pH = 7.4) but not in the Golgi (pH = 6.2).

2. Computational methods were used to calculate an interact strength between the Trp and the various K/R/H"-residues in the -4 position. However, computational analysis was not carried out to calculate the pKA of the K/R/H"-residues, i.e., such with PROPKA as I think constant pH MD simulations would be beyond the scope of this paper. Related to the above question, I think it would be important to show the relationship between the charged state of the -4 residues in the peptide with pH-dependent binding affinities.

We have calculated the -4 side chain pKa values for HDEL and KDEL, and this is described in our answer to main point 1 (see table). This is now described in the main text on pages 11-12, lines 300-309.

3. Related to the above 1 and 2 questions, is there any obvious difference with the proteins that use the RDEL sequence? For example, could these proteins be located in the Cis-Golgi wherein the pH difference is going to be less as compared to the pH of the TGN?

This question touches on what the properties of the different signals and the receptor mean for where in the Golgi proteins will be recognised. RDEL bearing proteins include the redox sensitive ER chaperone ERP44 and a group of sugar transferases for substrates such as collagen. Whether or not these proteins are more or less efficiently retrieved, and thus progress further into the Golgi is not known. One might expect HDEL proteins to retrieved first in the cis-Golgi since they bind with higher affinity, but as they are typically lower abundance this is probably an oversimplification.

4. It was unclear to me whether the KDEL receptor was sufficient to cope with the turnover of abundant proteins, e.g., like BIP. One could imagine that under ER-stress that other isoforms such as KDEL3, for example, is required to help cope with trafficking. Related to this question, is it known what the preferences are between the K/R/HDEL variants and the KDELR1-3 variants?

Although we have not included it in the current manuscript, we have looked at the efficiency of ER retrieval in cells using proteomics. In brief, that data shows all three receptors contribute to retrieval of abundant proteins such as BIP, PDI and calreticulin under normal conditions, so only a small fraction leaks from the cell. That is in line with previous studies and the original identification of ERD2 in yeast. Under stress conditions this may well change, and the different isoforms of the KDEL-receptor, specifically KDELR3, may be important under those circumstances. Some published work suggests that KDELR3 has a higher preference for HDEL, whereas KDELR1 and R2 have similar specificities. We have now examined the response of KDELR1 and R2 to H/K/R/A/DDEL ligands in cells (new Figure 1 – supplement 2). This confirms key aspects of our work relating to the specificity of KDELR1/2. We have also examined KDELR3 but found technical issues suggesting the tagged construct may not be trafficked normally. This limits our ability to say much more. KDELR3 is identical in the ligand binding pocket residues, but differs from KDELR1/2 in the loop before E117 important for selectivity. Whether or not this is important for any functions of KDELR3, possibly in ER stress isn’t known. This is now mentioned in the revised discussion. However, exploring these ideas will require detailed further work including new structures of KDELR3 beyond the scope of this study.

5. R169 is interesting to me as the residues is located at the end of TMs and will therefore snorkel into the membrane bilayer. Do you think that the lipid composition differences between the Golgi and ER (high in PE) could also contribute to fine-tuning signal binding?

We are currently exploring how lipid composition might influence the KDEL-receptor, and agree that R169 may be of potential interest in that context. However, it is likely to be complex, since the oligomeric state also needs to be taken into account. Whether or not there are differences for the three receptors isn’t known, but is an avenue for future work.

6. In my opinion, the terms "baton-relay" and "proof-reading" terminologies used imply more active processes than the simple biophysical interactions between the H/D/RDEL motif and the KDEL receptor.

Signal binding is of course mediated by simple biophysical interactions, however in the context of the KDEL-receptor these interactions drive a cycle of conformational change in the receptor. Crucially, our MD analysis shows that these interactions work together to effectively pull the retrieval signal into the receptor. Our data suggest this mechanism is more dynamic than a simple model of two objects colliding and sticking together. We accept that not everyone will agree with our terminology, but we felt it was helpful to explain what is happening during peptide binding to the receptor.

Nevertheless, we have revised the title and abstract to remove the term “baton-relay” and focus on the mechanism of signal capture and proofreading. In the discussion we use the more neutral term “relay” to describe the way the signal moves between arginine residues in the binding pocket. We note one of the reviewers describes the arginine residues as forming a “relay”, so this seems reasonable, and avoids the implication of a more active process implied by the term “baton-relay”. Proofreading is often a kinetic process largely dictated by chemical processes, or biophysical interactions of the kind we envisage are operating here. In many instances kinetic proofreading by enzymes is used to increase specificity, for example tRNA synthetases. For KDELR our data show ligand binding at mildy acidic pH triggers an allosteric change closing the binding pocket trapping the signal.

7. Could tryptophan fluorescence be used as a proxy for peptide binding? Such an assay would make it easier to understand how peptide binding and dynamics are coupled.

In principle we could use tryptophan fluorescence to study peptide binding. This is something we have not explored so far, but would allow us to better explore the dynamics of peptide binding in future.

8. Previous work has highlighted a role of D50 in the binding of the signal peptide, but the structures do not validate this claim, as the authors rightfully comment on. Still, closer inspection of the structures shows that D50 is at the same height as E117, which validated here to be an important residue, binding via ionic interactions to the positively charged residue of the signal sequences. Is it possible that the structures represent the end stage of the binding, a lower energy state. D50 could be another temporary entry point for the signal peptide, as stated in the text. If so, the mechanism could be equivalent to the insertion of a key, where the signal sequence binds to D50, but then rotates to E117 to lock the peptide in place, i.e. in the most stable binding pose, which could trigger the change in state leading to retrieval. The relay, made of three negatively charged groups on the signal sequence and three positively charged groups on the receptor, would still be important for driving the key into the lock. MD simulations can only probe a limited number of possible scenarios.

This is a very important point and the comment is extremely useful. We accept the MD simulations, although insightful, can only prove a limited number of potential scenarios. For that reason, we were rather cautious in our original text. The suggestion made here is reasonable and we have extended the discussion to cover this idea.

9. The authors note that they have used thermostability assays to study the folding of the receptor. My question is whether the binding of various signal peptides leads to differences in the thermostability of the receptor. Given the large number of ionic interactions involved in binding of the signal, it is certainly a possibility. It would provide a direct means to study the changes in stability of the receptor, which is related to aspects of energetics. If so, it might also be a route to study the interactions of the ER residents with the receptor.

We have to thank the reviewers for this helpful advice. The simple answer is yes, retrieval signal binding leads to differences in the thermostability of the receptor. These correlate well with signal affinity measured by other methods. We have used this approach to study RDEL binding to the receptor in comparison to HDEL and KDEL signals. This data was added as Figure 2 – supplement 1.